# M³Ret: A Mixed Multimodal Image Dataset and Benchmark for Personalized Multi-Retinal Disease Detection

## Abstract

In ophthalmic clinical practice, various imaging examinations, such as retinal fundus photography and OCT imaging, provide ophthalmologists with non-invasive methods to assess the condition of the retina and highlighting the importance of multimodal data. The imaging examinations are individually tailored according to each patient's clinical condition, resulting in diverse modality combinations. However, existing multimodal ophthalmic imaging datasets only collected one combination of multimodal data for single disease detection. Correspondingly, previous multimodal models were designed to learn from a fixed combination of modalities, overlooking the personalized nature of clinical examinations and the variability in modality combinations. As a result, the models often fail to generalize well to real-world clinical applications. To bridge the gap, this paper proposes (1) M³Ret, a **M**ixed **M**ultimodal ophthalmic imaging dataset for personalized **M**ulti-**Ret**inal disease detection, which consists of scanning laser ophthalmoscopy (SLO) images and optical coherence tomography (OCT) images and includes various modality combinations, and (2) PersonNet, a new baseline model for personalized multimodal multi-retinal disease detection, which can handle samples with various modality combinations during both training and inference phases, (3) benchmark results of our PersonNet and 13 existing multimodal learning methods, which demonstrate the superiority of the proposed PersonNet and highlight substantial room for improvement remains before clinical application can be achieved.

## 1 Introduction

2D retinal images and 3D Optical Coherence Tomography (OCT) provide ophthalmologists with non-invasive ways to assess the retinal fundus and screen for retinal diseases such as macular edema, diabetic retinopathy, age-related macular degeneration and glaucoma, and have been widely used in ophthalmology. To make the retinal disease screening automated and more efficient, several multimodal ophthalmic imaging datasets (Hassan et al., 2022) (Wang et al., 2022b) (Wu et al., 2023) (Luo et al., 2024b) (Luo et al., 2024a) have been released in recent years and significant progress has been made in multimodal learning to explore the complementary diagnostic information from the multimodal images for retinal disease detection.

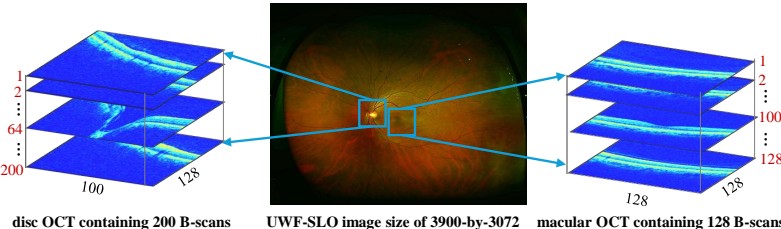

disc OCT containing 200 B-scans    UWF-SLO image size of 3900-by-3072    macular OCT containing 128 B-scans

Figure 1: An example for ultra wide field scanning laser ophthalmoscope (UWF-SLO) image, disc OCT images and macular OCT images.

Nevertheless, learning a multimodal retinal disease screening model for real-world clinical settings remains challenging due to the need for personalized examinations. The major challenges stem from the gap between recent datasets and real-world ophthalmic clinical applications, which we summarize as follows: (1) *Limited to samples with complete modality*: Most existing multimodal datasets assume complete modality availability for each sample, thereby neglecting the practical scenario of personalized imaging examinations, where samples may be either modality complete or modality incomplete, and the modality combinations differ among patients. Consequently, multimodal models (Zou et al., 2023; 2024; Wu et al., 2023) trained on modality complete samples lack flexibility and are unable to perform disease detection on modality incomplete samples. (2) *Limited to a single disease focus*: currently available datasets were collected targeted for single retinal disease detection and overlooked the coexistence of multiple diseases. For example, the latest dataset, FairVision (Luo et al., 2024a), consists of three subsets, each providing imaging of only a localized retinal structure associated with a single disease and lacking a broad overview of the retina which is necessary for detecting other coexisting retinal diseases. Consequently, models trained on this dataset are not applicable to detect other diseases. To bridge this gap, a new multimodal dataset and a corresponding learning solution that meet the practical needs of real-world clinical applications for personalized multi-retinal disease detection are highly desirable.

To mitigate the above two challenges, we introduce $M^3$Ret, a more real-world and large-scale multimodal dataset with seven modality combinations of high-resolution ultra wide field scanning laser ophthalmoscope (UWF-SLO) image, macular OCT images and disc OCT images (see Fig. 1) for the detection of three prevalent retinal diseases: diabetic retinopathy, diabetic macular edema and glaucoma. Compared to previous multimodal ophthalmic imaging datasets, our $M^3$Ret provides three distinct contributions to the community: (1) *Large-scale and multi-retinal diseases*: $M^3$Ret contains images from 8,558 individual eyes with labels for three prevalent retinal diseases and it is a multi-label multimodal dataset; (2) *Complementary views*: High-resolution UWF-SLO images and 3D OCT images focusing on two local anatomic structures—macula and optic disc—were collected. As shown in Fig. 1, the former provides a wide-field *en face* view of the retina, including optic disc, macular, peripheral lesions, vascular structures, and overall retinal morphology while the latter exhibits the cross-sectional views of the macula and optic disc, revealing detailed micro-structural changes such as retinal layer disruptions, and nerve fiber thinning etc. (3) *Mixed modality combinations*: Following practical clinical settings, $M^3$Ret collects both modality-complete samples and modality incomplete samples. In total, seven combinations of modalities are included.

With the new dataset, we propose a baseline method named PersonNet for personalized multi-retinal disease detection, which can handle both modality-complete and modality-incomplete samples during both training and inference. Specifically, we propose the personalized missing modality feature completion module which maintains a memory bank of modality-specific, class-wise prototypes and synthesizes the missing modality features by weighting the class-wise prototypes in the memory bank. Additionally, we propose the personalized fusion strategy to fuse the multimodal features to enhance the disease detection performances.

Finally, we propose a novel incomplete multimodal learning benchmark for personalized multi-retinal disease detection. The results demonstrate that the proposed PersonNet achieves the best while current state-of-the-art multimodal learning methods fail to achieve satisfactory performances. As $M^3$Ret is collected from real-world clinical practice, we hope that it could serve as a new benchmark for evaluating personalized multimodal multi-retinal disease detection and offer benefits to both the computer scientists and clinical ophthalmologists. In summary, the contributions of this work can be concluded as follows:

- We introduce $M^3$Ret, a **M**ixed **M**ultimodal ophthalmic imaging dataset for personalized multi-label **M**ulti-**Ret**inal disease detection. According to the real-world clinic settings, two tasks are defined. To the best of our knowledge, this is the first dataset that considers diverse modality combinations and supports personalized multi-retinal disease detection.

- We propose a strong multimodal learning baseline PersonNet for personalized multi-label multi-retinal disease detection which can adapt to various modality combinations.

- We benchmark current multimodal learning methods on $M^3$Ret with various evaluation metrics, revealing the limitations of existing state-of-the-art methods in addressing the missing modality problem.

## 2 RELATED WORK

**Multimodal ophthalmic datasets.** As summarized in Table 1, existing ophthalmic multimodal datasets are diverse in data modalities, scanned anatomical structures, number of individual eyes, disease types etc. For example, MMC-AMD (Wang et al., 2022b) and GAMMA (Wu et al., 2023) consider the two commonly used modalities in clinic: the 2D color fundus photograph (CFP) captured using traditional retinal cameras with field of views typically ranging from 30 to 60 degrees and macular OCT images. Although the CFP provides additional information about other anatomic structures such as disc and vessels, the sample sizes are small. Besides, MMC-AMD (Wang et al., 2022b) only collected one B-scan of the OCT images which miss abnormalities exhibited in other B-scans. Harvard-GDP (Luo et al., 2023) collects the 52-D vector of deviation values of visual fields and the 2D retinal nerve fiber layer thickness maps which are derived from the disc OCT images for glaucoma detection. Differently, Harvard-GF Luo et al. (2024b) collects the disc OCT images and derived the RNFLT as two modalities, together with the demographic information about patients for group fair glaucoma detection. However, Harvard-GDP (Luo et al., 2023) and Harvard-GF (Luo et al., 2024b) are specifically designed for glaucoma.

More recently, the largest multimodal ophthalmic imaging dataset, FairVision (Luo et al., 2024a), has been released. It contains disc/macular-centered SLO and OCT images which are simultaneously captured by the same device. FairVision (Luo et al., 2024a) includes three subsets for the detection of diabetic retinopathy (DR), age-related macular degeneration (AMD), and glaucoma, respectively. Although macular SLO and macular OCT provide complementary information about the macula, and disc SLO and disc OCT offer similar benefits for the optic disc, all of these modalities lack information about other anatomical structures in the retina. This limitation restricts their applicability to the detection of other retinal diseases.

**Multi-modal retinal disease detection.** Existing multimodal learning methods, such as MSAN (He et al., 2021) and the baseline method (Wu et al., 2023) adopt a two-branch architecture to learn modality-specific features from two different ophthalmic imaging modalities, and then directly concatenate the high-level modality-specific features for final disease detection. Instead, Wang et al. (2022a) and Wang et al. (2022a) fuse the final predictions from each branch via summation for AMD categorization, while Luo et al. (2023) concatenate the two modalities before feeding them into the single branch CNN network for glaucoma progression forecasting. Considering that the confidence of prediction by each modality is different, confidence-aware fusion strategy is proposed in EyeMoS$t$ (Zou et al., 2023) and EyeMoS$t$+ (Zou et al., 2024). Although these multimodal learning methods have achieved excellent performance in retinal disease detection, their application to real-world clinic scenarios is limited as they require modality complete samples as input and ignore the modality incomplete samples.

**Multimodal learning for modality missing inference.** Methods such as CorrKD (Li et al., 2024) and PCD (Chen et al., 2024) randomly dropout the modalities of modality complete samples to generate the modality incomplete samples, and then transfer the knowledge from the network trained with modality complete samples to the network taking modality incomplete samples as input. In this way, the model allows various modality combinations as inputs in the inference stage. Nevertheless, the modality incomplete samples are directly overlooked in the training stage.

**Incomplete multimodal learning with incomplete modality samples.** These methods aim to make use of modality complete samples and modality incomplete samples to train the multimodal models. Some try to complete the features for missing modalities. For example, RFNet (Ding et al., 2021) completes the features of the missing modality with zeros for tumor segmentation.

| Dataset | Diseases | #Devices | Modal 1 | Modal 2 | #Combinations | #Eyes | Year |
|---|---|---|---|---|---|---|---|
| MMC-AMD (Wang et al., 2022b) | AMD | 2 | CFP | Macular OCT$_{\times 1}$ | 2 | 1,093 | 2022 |
| GAMMA (Wu et al., 2023) | Glaucoma | 2 | CFP | Macular OCT$_{\times 256}$ | 1 | 300 | 2023 |
| Harvard-GDP (Luo et al., 2023) | Glaucoma | 2 | Visual Field (vector) | RNFLT | 1 | 1,000 | 2023 |
| Harvard-GF (Luo et al., 2024b) | Glaucoma | 1 | RNFLT | Disc OCT$_{\times 200}$ | 1 | 1,000 | 2023 |
| FairVision-DR (Luo et al., 2024a) | DR | 1 | Macular SLO | Macular OCT$_{\times 128}$ | 1 | 10,000 | 2024 |
| FairVision-AMD (Luo et al., 2024a) | AMD | 1 | Macular SLO | Macular OCT$_{\times 128}$ | 1 | 10,000 | 2024 |
| FairVision-GL (Luo et al., 2024a) | Glaucoma | 1 | Disc SLO | Disc OCT$_{\times 200}$ | 1 | 10,000 | 2024 |
| M³Ret (Ours) | DR & DME & Glaucoma | 2 | Ultra-widefield SLO | Macular OCT$_{\times 128}$ Disc OCT$_{\times 200}$ | 7 | 8,558 | 2025 |

Table 1: Comparison of multimodal ophthalmic datasets.

ShaSpec (Wang et al., 2023) decomposes the features of each modality into modality-shared features and modality-specific features, then complete the modality-shared features of the missing modality with the mean of the modality-shared features of available modalities. Similarly, MCKD (Wang et al., 2024) completes the features for missing modality with the mean of the available modality features, then learns the modality importance weight to fuse the available modality features and missing features within a meta-learning framework. To mitigate the completion for missing modalities, DMRNet (Wei et al., 2025) and IMDR (Liu et al., 2025) model the multimodal features of different modality combinations as a probabilistic distribution and sample the fused features from the distribution for classification while MLA (Zhang et al., 2024) learns a modality-specific encoder for each modality and a shared head for classification via alternating unimodal adaption and then integrates the predictions of available modalities with uncertainty based weights.

## 3 M³RET

### 3.1 DATASET CONSTRUCTION

**Dataset overview.** M³Ret includes two imaging modalities: 2D ultra-wide-field scanning laser ophthalmoscopy (UWF-SLO) images and 3D optical coherence tomography (OCT) images, designed for the detection of three prevalent retinal diseases: macular edema (ME), diabetic retinopathy (DR), and glaucoma. The data were collected from 8,558 individual eyes of 5,235 patients who visited the Ophthalmic Outpatient Department at *[Anonymization]* Hospital in between January 2019 and December 2022. All data and diagnostic reports were collected, anonymized, and stored securely. The study including data collection process, anonymization strategy, and storage protocol etc. were approved by the Medical Ethics Committee of *[Anonymization]*. All data are protected, and no personal information has been disclosed. Informed consent was waived due to the retrospective nature of the study.

**Data collection.** In M³Ret, UWF-SLO images were captured using the Optos Panoramic 200 scanning laser ophthalmoscope with resolutions of either $3900 \times 3072$ or $3072 \times 3072$ pixels. OCT images were acquired using the CIRRUS HD-OCT 500 device. Each macular OCT scan consists of 128 B-scans with a spatial resolution of $512 \times 1024$, covering a $6mm \times 6mm$ area centered on the macula. Each disc OCT scan consists of 200 B-scans with a spatial resolution of $200 \times 1024$, covering a $6mm \times 6mm$ area centered on the optic disc. In total, there are seven different modality combinations, and the number of samples and the proportion of each combination are summarized in Table 2. As shown, a total of 71.2% of individual eyes required only one type of examination (Uni-modal) and 27.4% required two (Bi-modal) while only 1.4% underwent three (Tri-modal) which reflects the fact that personalized nature of clinical assessments, where the extent of testing is tailored to the specific needs of each patient.

| #Modalities | Notations | Modality | | | #Eyes (Ratio) | |
|---|---|---|---|---|---|---|
| | | UWF-SLO | Macular OCT | Disc OCT | | |
| Uni-modal | Uni-1 | ✓ | | | 1,822 (30.0%) | |
| | Uni-2 | | ✓ | | 2,028 (23.7%) | 6,096 (71.2%) |
| | Uni-3 | | | ✓ | 2,246 (26.2%) | |
| Bi-modal | Bi-1 | ✓ | ✓ | | 1,228 (14.3%) | |
| | Bi-2 | ✓ | | ✓ | 1,052 (12.3%) | |
| | Bi-3 | | ✓ | ✓ | 64 (0.7%) | 2,344 (27.4%) |
| Tri-modal | Tri-1 | ✓ | ✓ | ✓ | 118 (1.4%) | 118 (1.4%) |
| **#Eyes in each modality** | | **4,220** | **3,438** | **3,480** | **Total: 8,558 (100%)** | |

Table 2: Distribution of eye imaging modality combinations and ratios.

**Labeling.** For each individual eye sample, labels for ME, DR and glaucoma were determined via retrieving the electronic medical record system. Labels for ME and DR are binary and the label for glaucoma is either "glaucoma", "non-glaucoma" or "suspicious". For samples with recorded diagnosis decisions, we directly assigned labels according to the diagnosis decisions. For samples whose diagnosis decisions were missing in the system but the detailed medical treatments were recorded, the disease labels were determined by experienced ophthalmologists according to the treatment records. Otherwise, the disease labels were marked as "unclear".

| Sub-dataset | Labels | Modality | Sources | #Eyes | Train | Val | Test |
|---|---|---|---|---|---|---|---|
| $\mathcal{D}_A$ | ME, DR, Glaucoma | UWF-SLO | Uni-1 & Bi-2 | 2,874 | 1,722 | 576 | 576 |
| | | Macular OCT | Uni-2 & Bi-3 | 2,092 | 1,216 | 406 | 470 |
| | | UWF-SLO & Macular OCT | Bi-1 & Tri-1 | 1,346 | 808 | 269 | 269 |
| | | **Total** | | **6,312** | **3,746** | **1,251** | **1,315** |
| $\mathcal{D}_B$ | Glaucoma | UWF-SLO | Uni-1 & Bi-1 | 3,050 | 1,830 | 610 | 610 |
| | | Disc OCT | Uni-3 & Bi-3 | 2,310 | 1,344 | 451 | 515 |
| | | UWF-SLO & Disc OCT | Bi-2 & Tri-1 | 1,170 | 700 | 235 | 235 |
| | | **Total** | | **6,530** | **3,874** | **1,296** | **1,360** |

Table 3: Summary of $\mathcal{D}_A$ and $\mathcal{D}_B$ including sources and data splits for training, validation and test.

**Task description and data splits.** In clinical practice, UWF-SLO images, macular OCT, and their paired combinations are commonly used to diagnose macular edema (ME), diabetic retinopathy (DR), and glaucoma. Specifically, UWF-SLO images and paired UWF-SLO with disc OCT images are primarily used for diagnosing glaucoma. However, only a limited number of patients undergo all three examinations or both macular and disc OCT scans, as shown in Table 2. Accordingly, **we define two tasks based on available modality combinations**: (1) three-disease detection (ME, DR, and glaucoma) using three combinations of UWF-SLO and macular OCT images, and (2) glaucoma detection using three combinations of UWF-SLO and disc OCT images. To fully use of the data for these tasks, we reorganize it to two sub-datasets as follows:

- $\mathcal{D}_A$ used for the detection of ME, DR and glaucoma with following three modality combinations: (1) UWF-SLO images from subsets *Uni-1* and *Bi-2*, (2) macular OCT images in *Uni-2* and *Bi-3*, and (3) paired UWF-SLO and macular OCT images from *Bi-1* and *Tri-3*. A total of 6,312 individual eyes are included in this dataset.
- $\mathcal{D}_B$ for glaucoma detection with three modality combinations: (1) UWF-SLO images from subsets *Uni-1* and *Bi-1*, (2) disc OCT images in *Uni-3* and *Bi-3*, and (3) paired UWF-SLO and disc OCT images from *Bi-2* and *Tri-3*. Totally, 6530 individual eyes are included.

We adopt stratified sampling strategy to split $\mathcal{D}_A$ and $\mathcal{D}_B$ into training, validation, and test sets with an approximate ratio of 6:2:2 and the number of eyes in each subset is shown in Table 3.

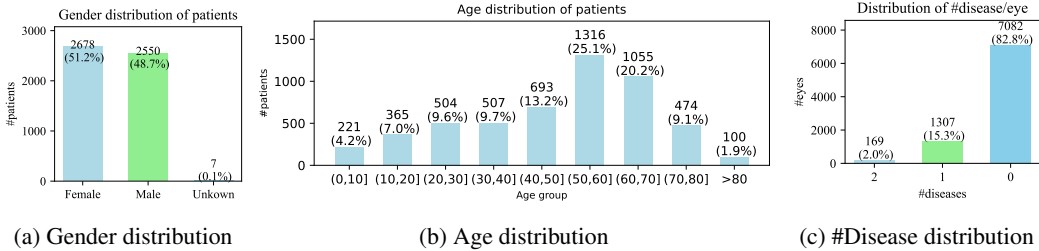

(a) Gender distribution      (b) Age distribution      (c) #Disease distribution

Figure 2: The distributions of gender and age across a cohort of 5,235 patients and distributions of number of diseases per eye across the 8558 eyes.

## 3.2 DATASET STATISTICS

**Cohort statistics.** To analysis the characteristics of M³Ret, we summarize the gender and age distributions of the 5235 patients in Fig. 2a and Fig. 2b respectively. Additionally, we report the distribution of the eyes suffering the number of diseases in Fig. 2c. As is shown, 2.0% of eyes suffer from two diseases while 15.3% of eyes suffer from one disease which indicates that a considerable number of eyes (2/17.3) have two diseases and they should not be ignored. The disease classification statistics for patients and eyes are list in Table 4. We observe a rate of 4.59% (240/5235) for ME and 7.07% (370/5235) for DR in our M³Ret dataset, which are close to the reported prevalence rates of 4.07% for ME and 6.17% for DR by Teo et al. (2021). The rate of glaucoma is 9.61% (503/5235), which is also comparable to the reported prevalence rate of 10.12% among the Chinese population in US (Stein et al., 2011). In contrast, FairVision (Luo et al., 2024a) reports a glaucoma rate of 48.7%, which deviates significantly from the prevalence rate reported by Stein et al. (2011). These comparisons clearly indicate that our data are derived from real-world clinical practice.

| | ME | | | DR | | | Glaucoma | | | | Total |
|---|---|---|---|---|---|---|---|---|---|---|---|
| | ME | non-ME | unclear | DR | non-DR | unclear | glaucoma | suspicious | non-glaucoma | unclear | |
| patients | 240 | 4993 | 2 | 370 | 4861 | 4 | 503 | 161 | 4350 | 221 | 5235 |
| eyes | 301 | 8253 | 4 | 589 | 7964 | 5 | 755 | 301 | 7076 | 426 | 8253 |

Table 4: Disease classification statistics for patients and eyes

| Modality | Total | ME | | | DR | | | Glaucoma | | | |
|---|---|---|---|---|---|---|---|---|---|---|---|
| | | ME | non-ME | unclear | DR | non-DR | unclear | glaucoma | suspicious | non-glaucoma | unclear |
| UWF-SLO | 2874 | 65 | 2807 | 2 | 189 | 2684 | 1 | 190 | 131 | 2433 | 120 |
| Macular OCT | 2092 | 130 | 1960 | 2 | 161 | 1931 | 0 | 130 | 35 | 1861 | 66 |
| UWF-SLO & Macular OCT | 1346 | 104 | 1242 | 0 | 225 | 1117 | 4 | 34 | 13 | 1281 | 18 |
| **Total** | 6312 | 299 | 6009 | 4 | 575 | 5732 | 5 | 354 | 179 | 5575 | 204 |

Table 5: Classification counts for ME, DR, and Glaucoma on $\mathcal{D}_A$

| Modality | Total | Glaucoma | Suspicious | Non-glaucoma | Unclear |
|---|---|---|---|---|---|
| UWF-SLO | 3050 | 53 | 5 | 2976 | 16 |
| Disc OCT | 2310 | 416 | 127 | 1543 | 224 |
| UWF-SLO & Disc OCT | 1170 | 171 | 139 | 738 | 122 |
| **Total** | 6530 | 640 | 271 | 5257 | 362 |

Table 6: Glaucoma classification counts on $\mathcal{D}_B$.

**Disease statistics in $\mathcal{D}_A$ and $\mathcal{D}_B$.** We present the disease statistics for $\mathcal{D}_A$ and $\mathcal{D}_B$ in Table 5 and Table 6, respectively. From these tables, we observe that the three diseases naturally exhibit a long-tail distribution. The disease distribution of the training, validation and test sets of $\mathcal{D}_A$ and $\mathcal{D}_B$ can be found in the supplementary materials.

**Evaluation metrics.** We follow the previous studies (Wu et al., 2023; Hu et al., 2024) and use the accuracy ($Acc$) and Cohen's Kappa (McHugh, 2012) to evaluate the detection performance for each disease. As the imbalanced class distribution in collected dataset, F1-score ($F1$), as the harmonic mean of specificity and sensitivity, is used for two-class classification of ME and DR and macro-F1 (Opitz & Burst, 2021) is applied for three-class glaucoma detection. For overall evaluation, the mean accuracy ($mAcc$), mean Cohen's Kappa ($mKappa$) and mean F1-score ($mF1$) across diseases are adopted. Additionally, considering the computation efficiency, we suggest that future work report the GPU memory usage, the number of parameters for evaluating model capacity, computational costs in FLOPS, and inference speed (FPS).

## 4 PERSONNET: PERSONALIZED MULTIMODAL MULTI-DISEASE DETECTION

**Framework.** Our PersonNet adopts a two-branch multimodal learning framework with personalized missing modality completion module and personalized multimodal feature fusion module for personalized multi-retinal disease detection, as shown in Fig. 3. In detail, we employ ResNet-50 (He et al., 2016) pre-trained on ImageNet-21K as the encoder for 2D UWF-SLO modality and 3D ResNet-50 (Hara et al., 2018) pre-trained on Kinetics-700, Moments-in-Time and STAIR-Actions as the encoder for 3D OCT modality. To capture the spatial contexts, the SE module (Hu et al., 2018) is inserted after the 4-th stage in both encoders. For modality complete samples, with the 2D ResNet and 3D ResNet, we can obtain their features directly, and fuse them for multi-disease detection as shown

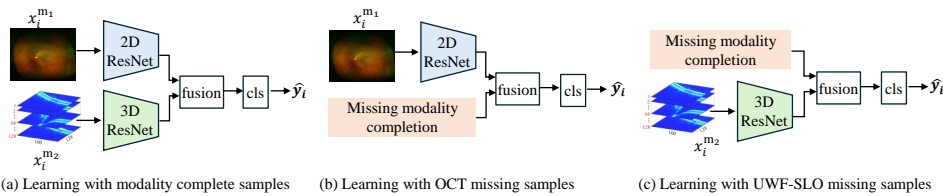

(a) Learning with modality complete samples     (b) Learning with OCT missing samples     (c) Learning with UWF-SLO missing samples

Figure 3: Framework of PersonNet for personalized multimodal multi-retinal disease detection.

in Fig. 3(a). For modality incomplete samples, missing modality completion module is required, then we fuse the completed features and features of the exist modality for multi-disease detection as shown in Fig. 3(b) and (c). The weights of the two encoders and classifiers are shared across the three modality combinations. To address the long-tailed distribution of diseases, we adopt a class-balanced cross-entropy loss:

$$\mathcal{L} = -\sum_{t=1}^{T} \sum_{c_t=1}^{C_t} \frac{1 - \gamma_{t,c_t}}{C_t - 1} \cdot y_{i,t,c_t} \cdot \log(\hat{y}_{i,t,c_t}) \,, \tag{1}$$

where $C_t$ is the number of categories of the $t$-th disease and $\gamma_{t,c_t}$ is the ratio of class $c_t$ for disease $t$, and $y_{i,t,c_t} = 1$ if the ground-truth label of sample $i$ regarding disease $t$ is $c_t$ otherwise $y_{i,t,c_t} = 0$, and $\hat{y}_{i,t,c_t}$ is the predicted probability belonging to $c_t$ regarding disease $t$.

**Personalized missing modality completion.** The most intuitive way is to directly complete the features of the missing modality with zeros so that existing multimodal learning methods are applicable. However, zero completion ignores the correlation between two modalities. Thus, we propose personalized missing modality completion module. It maintains a memory bank of class-wise prototypes for each modality and measuring the similarities between the features of the existing modality and the class-wise prototypes of the missing modality. Then, these similarities are used as importance weights to synthesize the features of the missing modality. The technical details can be found in the supplementary materials.

**Personalized multimodal feature fusion.** Obviously, the modality-specific features extracted directly from the input are more discriminative than those synthesized with the prototypes. Thus, it is desired to fuse modality-specific features from samples with various modality combinations in different ways. To this end, we propose to separately learn the important weight of each modality-specific features for each modality combination in a similar way to SKNet (Li et al., 2019), then fuse them for multi-disease classification. The technical details can be found in the supplementary materials.

## 5 EXPERIMENT

### 5.1 EXPERIMENTAL SETUPS

**Pre-processing and data augmentation.** For UWF-SLO images, we rescale them with the short side of 512. Then, we keep the long side be 640 via random cropping or zero padding for images with longe sides greater or less than 640. Additionally, we apply random rotations within the range of $-30°$ to $30°$, random horizontal flipping and brightness enhancement within the range of 0-0.9 to augment the training set. For OCT modality, we downsample macular OCT images to $128 \times 128 \times 128$ and disc OCT images to $200 \times 100 \times 128$, and then pad disc OCT with zeros to $200 \times 128 \times 128$. To augment OCT data, padding and random cropping without altering the data size is applied. Additionally, random rotation within the range of $-15°$ to $15°$, random horizontal flipping, and scan-wise duplication or discarding are performed.

**Implementation details.** We conduct experiments on MMPreTrain platform (Contributors, 2023). SGD optimizer is adopted to train the multimodal models for 150 epochs, with the learning rate decaying by a factor of 0.1 at the 120-epoch milestone. Other hyper-parameters include an initial learning rate of $8 \times 10^{-3}$ and a batch size of 16. All experiments are conducted using a single NVIDIA GeForce RTX 3090 GPU with 24 GB of memory.

**Baselines for complete multimodal learning**. Addition to the baseline for personalized multimodal learning method we propose, we select four vanilla multimodal learning methods and four recent state-of-the-art (SOTA) multimodal learning methods with zero completion for the missing modality as baseline methods. The four vanilla multimodal learning methods include: (1) Combination specific detector (MultiModel) which separately trains a disease detection model for each modality combination, and (2) three baseline multimodal fusion methods: feature summation (Sum), feature concatenation (Concat) and late fusion by logits summation (LateFusion). The four SOTAs are FiLM (Perez et al., 2018), BiGated (Kiela et al., 2018), MMCNN (Wang et al., 2019),and LFM (Yang et al., 2024). For fair comparisons, all methods use ResNet-50 as the SLO encoder and 3D ResNet-50 as the OCT encoder.

**Baselines for incomplete multimodal learning**. We select five incomplete multimodal methods which support both modality complete samples and modality incomplete samples. They are LCR (Zhou et al., 2020), MMANet (Wei et al., 2023), ShaSpec (Wang et al., 2023), MLA (Zhang et al., 2024) and DMRNet (Wei et al., 2025). For fair comparisons, all methods use ResNet-50 as the SLO encoder and 3D ResNet-50 as the OCT encoder.

## 5.2 BENCHMARK RESULTS ON $\mathcal{D}_A$ FOR MULTI-RETINAL DISEASE DETECTION

**Results of complete multimodal learning methods.** We train multi-retinal disease detection models on the training set of $\mathcal{D}_A$ and select the optimal hyperparameters according to the performances on validation set and report the performances on test set. We first evaluate the performances of our PersonNet against the compared baselines of complete multimodal methods on $\mathcal{D}_A$ and report them in Tab. 7. Additionally, the GPU memory consumption, the number of parameters, the floating-point operations per second (FLOPs) and inference speed (FPS) are also reported. Undoubtedly, equipped with our missing modality completion module and personalized fusion module, the proposed personalized disease detector consistently outperforms MultiModel and seven other baseline multimodal learning methods by a considerable margin, achieving the highest $mAcc$ of 93.38%, $mF1$ of 57.03%, and $mKappa$ of 49.21%, surpassing the second-best method by 0.01%, 1.96%, and 2.15%, respectively. However, the recent SOTA method LFM (Yang et al., 2024), which integrates unsupervised contrastive learning to align multimodal features and alleviate the modality imbalance problem, performs the worst in terms of $mF1$ and $mKappa$. A possible reason is that exploiting complementary semantics from multimodal data is more effective for disease detection rather than aligning the modality-specific features to the same embedding space. Compared to the MultiModel which trains three models separately for each modality combination, multimodal learning methods except for LFM (Yang et al., 2024) demonstrate superior performance in terms of $mF1$ and $mKappa$, revealing that exploiting complementary enhances disease detection performances.

**Results of incomplete multimodal learning methods.** We report the performance of five incomplete multimodal learning methods in Tab. 7. As shown, the proposed baseline PersonNet outperforms the second-best method by 0.18% in $mAcc$, 1.80% in $mF1$, and 1.72% in $mKappa$, respectively. Although these methods attempt to address the missing modality problem, they do not yield significant performance improvements over the naive multimodal learning approach, which simply fills in the missing modality with zeros. Future research in incomplete multimodal learning should focus on more effective solutions for handling missing modalities and on personalized feature fusion strategies to achieve better detection performance.

| Category | Method | mAcc | mF1 | mKappa | GPU Mem (MB) | #Params (M) | FLOPs | FPS (sample/s) |
|---|---|---|---|---|---|---|---|---|
| Combination specific (MultiModel) | | $93.37_{\pm0.13}$ | $52.13_{\pm0.97}$ | $43.36_{\pm1.23}$ | 440.89 | 69.69 | 27.34 | 4.91 |
| Complete multimodal learning | Concat | $92.68_{\pm0.33}$ | $54.67_{\pm0.85}$ | $46.42_{\pm1.00}$ | 440.89 | 69.69 | 27.34 | 4.91 |
| | Sum | $92.39_{\pm0.45}$ | $52.94_{\pm0.46}$ | $45.07_{\pm1.27}$ | 440.84 | 69.68 | 27.34 | 4.98 |
| | LateFusion | $92.48_{\pm0.21}$ | $55.07_{\pm0.84}$ | $47.06_{\pm1.05}$ | 440.89 | 69.69 | 27.34 | 4.85 |
| | FiLM (Perez et al., 2018) | $93.07_{\pm0.40}$ | $54.88_{\pm1.05}$ | $46.94_{\pm1.32}$ | 472.85 | 78.07 | 27.35 | 4.91 |
| | BiGated (Kiela et al., 2018) | $92.58_{\pm0.33}$ | $54.29_{\pm1.15}$ | $46.09_{\pm1.40}$ | 472.85 | 78.07 | 27.35 | 4.84 |
| | MMCNN (Wang et al., 2019) | $92.96_{\pm0.42}$ | $53.84_{\pm1.09}$ | $46.00_{\pm1.17}$ | 472.91 | 78.08 | 27.35 | 4.95 |
| | LFM (Yang et al., 2024) | $93.26_{\pm0.12}$ | $50.99_{\pm2.14}$ | $42.17_{\pm2.12}$ | 440.89 | 69.69 | 27.34 | 4.74 |
| | PersonNet(**Ours**) | $\mathbf{93.38_{\pm0.30}}$ | $\mathbf{57.03_{\pm0.96}}$ | $\mathbf{49.21_{\pm1.23}}$ | 462.06 | 75.21 | 27.35 | 4.75 |
| Incomplete multimodal learning | LCR (Zhou et al., 2020) | $93.20_{\pm0.18}$ | $54.76_{\pm0.71}$ | $46.39_{\pm0.90}$ | 584.96 | 107.46 | 27.38 | 4.74 |
| | MMANet (Wei et al., 2023) | $92.51_{\pm0.22}$ | $54.13_{\pm1.13}$ | $45.64_{\pm1.26}$ | 717.72 | 69.69 | 54.68 | 4.67 |
| | ShaSpec (Wang et al., 2023) | $90.07_{\pm0.42}$ | $49.99_{\pm1.26}$ | $40.83_{\pm1.57}$ | 516.98 | 88.41 | 66.20 | 4.55 |
| | MLA (Zhang et al., 2024) | $93.16_{\pm0.30}$ | $55.23_{\pm1.44}$ | $47.49_{\pm1.43}$ | 440.84 | 69.68 | 27.34 | 4.61 |
| | DMRNet (Wei et al., 2025) | $90.31_{\pm1.38}$ | $51.04_{\pm1.05}$ | $41.81_{\pm0.82}$ | 472.85 | 78.09 | 27.35 | 4.81 |
| | PersonNet(**Ours**) | $\mathbf{93.38_{\pm0.30}}$ | $\mathbf{57.03_{\pm0.96}}$ | $\mathbf{49.21_{\pm1.23}}$ | 462.06 | 75.21 | 27.35 | 4.75 |

Table 7: Overall performances of the proposed PersonNet and 13 baseline methods on $\mathcal{D}_A$ for multi-retinal disease detection. Means and standard deviations over five trials are reported.

## 5.3 BENCHMARK RESULTS ON $\mathcal{D}_B$ FOR GLAUCOMA DETECTION

**Results of complete multimodal learning methods.** Similarly, we train the complete multimodal learning models for glaucoma detection on the training set of $\mathcal{D}_B$ and select the optimal hyperparameters according to the performances on the validation set and report the detection performances on

| Category | Method | mAcc | mF1 | mKappa | GPU Mem (MB) | #Params (M) | FLOPs | FPS (Img/s) |
|---|---|---|---|---|---|---|---|---|
| Combination specific (MultiModel) | | $87.63_{\pm0.40}$ | $60.19_{\pm2.10}$ | $51.17_{\pm1.57}$ | 502.48 | 69.69 | 27.55 | 4.46 |
| Complete multimodal learning | Concat | $88.38_{\pm0.80}$ | $62.50_{\pm1.80}$ | $55.38_{\pm2.06}$ | 502.48 | 69.69 | 27.55 | 4.46 |
| | Sum | $87.44_{\pm0.46}$ | $58.53_{\pm0.70}$ | $52.73_{\pm1.55}$ | 502.42 | 69.68 | 27.55 | 4.38 |
| | LateFusion | $87.82_{\pm0.26}$ | $59.47_{\pm1.49}$ | $53.02_{\pm1.81}$ | 502.48 | 69.69 | 27.55 | 4.39 |
| | FiLM (Perez et al., 2018) | $88.54_{\pm0.22}$ | $62.56_{\pm1.50}$ | $55.99_{\pm0.90}$ | 534.44 | 78.07 | 27.55 | 4.24 |
| | BiGated (Kiela et al., 2018) | $87.61_{\pm0.76}$ | $61.22_{\pm2.20}$ | $55.23_{\pm1.11}$ | 534.44 | 78.07 | 27.55 | 4.15 |
| | MMCNN (Wang et al., 2019) | $88.44_{\pm0.69}$ | $61.66_{\pm2.10}$ | $53.85_{\pm2.29}$ | 534.49 | 78.08 | 27.55 | 4.46 |
| | LFM (Yang et al., 2024) | $88.97_{\pm0.51}$ | $59.31_{\pm3.84}$ | $52.05_{\pm3.07}$ | 502.48 | 69.69 | 27.55 | 4.26 |
| | PersonNet(Ours) | $89.12_{\pm0.45}$ | $65.60_{\pm1.09}$ | $58.09_{\pm2.62}$ | 523.64 | 75.21 | 27.56 | 4.23 |
| Incomplete multimodal learning | LCR (Zhou et al., 2020) | $88.67_{\pm0.53}$ | $63.36_{\pm1.94}$ | $54.83_{\pm1.98}$ | 646.54 | 107.16 | 27.58 | 4.16 |
| | MMANet (Wei et al., 2023) | $88.08_{\pm0.44}$ | $60.66_{\pm2.14}$ | $53.73_{\pm1.55}$ | 779.68 | 69.69 | 55.09 | 4.10 |
| | ShaSpec (Wang et al., 2023) | $86.08_{\pm1.37}$ | $60.18_{\pm1.50}$ | $52.38_{\pm3.11}$ | 522.06 | 88.43 | 66.23 | 4.13 |
| | MLA (Zhang et al., 2024) | $87.49_{\pm0.45}$ | $57.92_{\pm1.30}$ | $49.46_{\pm1.28}$ | 502.42 | 69.68 | 27.55 | 4.15 |
| | DMRNet (Wei et al., 2025) | $87.26_{\pm1.02}$ | $60.72_{\pm1.40}$ | $55.03_{\pm2.36}$ | 534.43 | 78.09 | 27.55 | 4.24 |
| | PersonNet(Ours) | $89.12_{\pm0.45}$ | $65.60_{\pm1.09}$ | $58.09_{\pm2.62}$ | 523.64 | 75.21 | 27.56 | 4.23 |

Table 8: Overall performances of the proposed PersonNet and 13 baseline methods on $\mathcal{D}_B$ for glaucoma detection. Means and standard deviations over five trials are reported.

test set in Tab. 8. As shown, our PersonNet achieves the best performances with $mAcc$ of 89.12%, $mF1$ of 65.60%, and $mKappa$ of 58.09%, surpassing the second-best by 0.15%, 3.04% and 2.10%, respectively. Similarly, we observe that the latest complete multimodal learning method does not exhibits its superiority in glaucoma detection in $mF1$ and $mKappa$ compared to the most naive baselines e.g. Concat and LateFusing while fusion the modality-specific features via Feature-wise Linear Modulation, i.e, FiLM (Perez et al., 2018) enhances the detection performances, achieving the second best in terms of $mF1$ and $mKappa$. This also demonstrates that exploiting complementary information from multimodal data e.g. FiLM (Perez et al., 2018) leads to better performances than aligning the modality-specific features like LFM (Yang et al., 2024).

**Results incomplete multimodal learning methods.** We then report the performances of the five incomplete multimodal learning methods in Tab. 8. As shown, our PersonNet surpasses the second-best by 0.45% in $mAcc$, 2.24% in $mF1$ and 3.06% in $mKappa$. Similarly, for the other five baselines, we observe that they do not exhibit superiority to most of the complete multimodal learning methods with zero completion for missing modality. This reveals that glaucoma detection performance on $\mathcal{D}_B$ still has significant room for improvement, and novel methods addressing missing modality completion and personalized feature fusion are encouraged to enhance detection performance.

## 5.4 Limitations

**Far from reaching a satisfactory level of agreement with clinical diagnosis.** To further analysis the disease detection performances of our PersonNet and baselines of both complete and incomplete multimodal learning methods, we report the performances of each disease in the supplementary. Although PersonNet achieves better performance than the compared methods, there is still substantial room for improvement to reach a satisfactory level of agreement which requires $Kappa \in [0.81, 1.00]$ for clinical applicability. In detail, the $kappa$ by PersonNet on $\mathcal{D}_A$ for ME, DR and glaucoma are 48.42%, 66.13% and 33.08% respectively (see supplementary for details ) while the glaucoma detection on $\mathcal{D}_B$ is 58.09%. According to the guidelines for the strength of agreement indicated with $Kappa$ (Landis & Koch, 1977; Kundel & Polansky, 2003), only the detection of DR on $\mathcal{D}_A$ reaches to the level of substantial agreement ($Kappa \in [0.61, 0.80]$) while the detection of ME on $\mathcal{D}_A$ and glaucoma on $\mathcal{D}_B$ reaches to the level of moderate agreement ($Kappa \in [0.41, 0.60]$) and the detection of glaucoma on $\mathcal{D}_A$ only reaches to the level of fair agreement ($Kappa \in [0.21, 0.40]$). Thus, numerous efforts are needed to improve the performances to reach a satisfactory level of agreement.

## 6 Conclusion

We introduce (1) M³Ret, the first dataset that considers diverse modality combinations, and (2) PersonNet a strong baseline which can adapt to various modality combinations, and (3) a benchmark for personalized disease detection. Benchmark results show that our PersonNet achieves best performances in personalized multi-retinal disease detection and there exists a large room to reach a satisfactory level of agreement with clinical diagnosis for retinal disease detection.

**Reproducibility Statement.** The dataset can be found at `https://drive.google.com/drive/folders/1mskJMpOQC-a2PAPXIHsAc1j4uOFUUWpV?usp=drive_link` and code can be found at `https://anonymous.4open.science/r/PersonNet-C12A/README.md`.

**Ethics Statement.** This study including data collection process, anonymization strategy, and storage protocol, data sharing protocol etc. was approved by the Medical Ethics Committee of *[Anonymization]*.

**The Use of LLM**. LLM is used to polish writing and generate the latex codes for tables in this submission.

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

# A  APPENDIX

## A.1  DISEASE STATISTICS IN $\mathcal{D}_A$ AND $\mathcal{D}_B$

To make sure the disease class distributions of the training, validation and test sets are same, stratified sampling strategy is adopted to split $\mathcal{D}_A$ and $\mathcal{D}_B$. In detail, we group the samples with same diseases. For each group, we divide it into three subsets with the ratio of 6:2:2 and merge the samples in three subsets into training, validation, and test sets respectively. In this way, the disease class distributions are enforced to be similar. Tab. 9 and Tab. 10 show the class distributions of each subset of $\mathcal{D}_A$ and $\mathcal{D}_B$ respectively.

| Split | ME | | | DR | | | Glaucoma | | | |
|---|---|---|---|---|---|---|---|---|---|---|
| | **ME** | **non-ME** | **unclear** | **DR** | **non-DR** | **unclear** | **glaucoma** | **suspicious** | **non-glaucoma** | **unclear** |
| Train | 181 | 3561 | 4 | 346 | 3395 | 5 | 201 | 102 | 3311 | 132 |
| Validation | 59 | 1192 | 0 | 114 | 1137 | 0 | 69 | 36 | 1111 | 35 |
| Test | 59 | 1256 | 0 | 115 | 1200 | 0 | 84 | 41 | 1153 | 37 |
| Total | 299 | 6009 | 4 | 575 | 5732 | 5 | 354 | 179 | 5575 | 204 |

Table 9: Number of eyes per disease label across data splits in $\mathcal{D}_A$.

| Split | Glaucoma | Suspicious | Non-glaucoma | Unclear |
|---|---|---|---|---|
| Train | 371 | 156 | 3123 | 224 |
| Validation | 127 | 55 | 1046 | 68 |
| Test | 142 | 60 | 1088 | 70 |
| **Total** | 640 | 271 | 5257 | 362 |

Table 10: Number of eyes for each glaucoma-related label across splits in $\mathcal{D}_B$.

## A.2 TECHNICAL DETAILS ABOUT PersonNet

In this subsection, we will first give the problem formulation for the setting of the personalized multimodal disease detection, then present the technical details about the newly designed incomplete multimodal learning framework in particularly the two key modules: **Personalized Missing Modality Completion** and **Personalized Multimodal Feature Fusion**.

### A.2.1 PROBLEM FORMULATION

Formally, suppose we are given the training set $\mathcal{D} = \left( x_i^{m_1} \cdot \delta_i^{m_1}, x_i^{m_2} \cdot \delta_i^{m_2}; y_i \right)_{i=1}^{N}$ where $x_i^{m_1}$ and $x_i^{m_2}$ represent two different modalities, e.g., UEF-SLO image and OCT image, of the $i-$th sample, and $\delta_i^{m_1} \in \{0, 1\}$ and $\delta_i^{m_2} \in \{0, 1\}$ indicate whether the modality is available. Here, $y_i = \{y_{i,t}\}_{t=1}^{T}$, $T$ is the number of diseases and $y_{i,t}$ is the ground-truth labels for $t-$th diseases. We note that $\delta_i^{m_1} = \delta_i^{m_2} = 1$ if the sample is modality complete; $\delta_i^{m_1}$ or $\delta_i^{m_2}$ is set to 0 if $m_1$ or $m_2$ is missing. Our goal is to train a multimodal model with $\mathcal{D}$ for disease detection so that the model is applicable to real-world samples with various modality combinations.

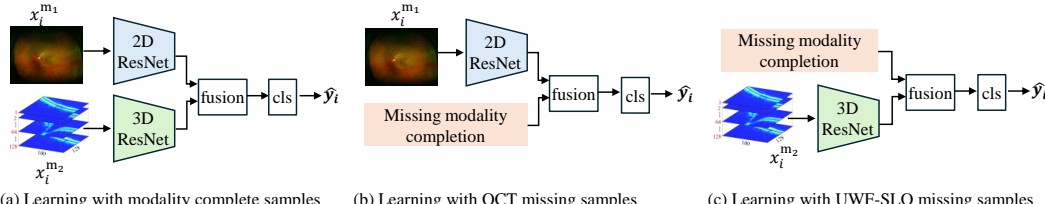

(a) Learning with modality complete samples    (b) Learning with OCT missing samples    (c) Learning with UWF-SLO missing samples

Figure 4: Framework of PersonNet for personalized multimodal multi-retinal disease detection. The 2D ResNets in (a) and (b) share the same parameters and the 3D ResNets in (b) and (c) share the same parameters.

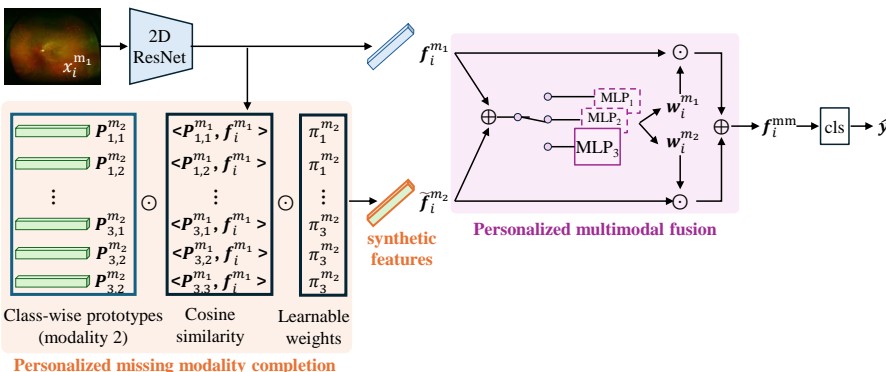

Figure 5: The pipeline of PersonNet learning with OCT missing samples.

### A.2.2 FRAMEWORK

**Overview.** To solve the missing modality problem and learn with various modality combinations, we propose PersonNet. Its framework is illustrated in Fig. 4. PersonNet consists of two modality-specific encoders, i.e., 2D ResNet and 3D ResNet for modality-specific feature extraction, two personalized missing modality completion modules and one personalized feature fusion module to fuse modality-specific features for disease detection. Fig. 4(a) illustrates the pipeline for modality complete samples, and Fig. 4(b) and (c) illustrates the pipeline for modality missing samples. For three various modality combinations, the parameters within encoders and personalized multimodal fusion module are shared.

**Modality-specific prototype maintenance.** For each modality, we maintain a memory bank of modality-specific, class-wise prototypes. We denote the memory bank for modality $m \in \{m_1, m_2\}$ as $\mathbf{P}^m = \{\mathbf{P}_t^m\}_{t=1}^T$ where $\mathbf{P}_t^m$ is the set of prototypes for disease $t$. Suppose the $t$-th disease has $C_t$ classes, then $\mathbf{P}_t^m = \{\mathbf{P}_{t,c_t}^m\}_{c_t=1}^{C_t}$. We dynamically maintain the prototypes during training. In detail, for each sample $i$ within one batch, with the modality-specific features via $\mathbf{f}_i^m = E^m(\mathbf{x}_i^m)$ where $m \in \{m_1, m_2\}$, we update the prototype $\mathbf{P}_{t,c_t}^m$ regarding disease $t$ and class $c_t$ via:

$$\mathbf{P}_{t,c_t}^m = \alpha \cdot \mathbf{P}_{t,c_t}^m + (1 - \alpha) \cdot \frac{1}{n_{c_t}} \sum_i \mathbf{f}_i^m \cdot \mathbf{1}(y_{i,t} = c_t), \tag{2}$$

where $\alpha$ represents the update weights, which increase linearly as training progresses through the warm-up epochs, and $\mathbf{1}(y_{i,t} = c_t)$ is an indicator function which returns 1 if the sample belongs to class $c_t$ regarding disease $t$ otherwise returns 0, and $n_{c_t}$ denotes the number of samples belonging to class $c_t$ regarding disease $t$ within the batch.

In what follows, we illustrate the detailed pipeline for addressing the samples without OCT modality, i.e., $m_2$ which is shown in Fig. 5. The pipeline for addressing samples without UWF-SLO image modality, i.e., $m_1$ is similar.

### A.2.3 PERSONALIZED MISSING MODALITY COMPLETION

**Similarities between feature and prototypes.** For a given modality incomplete sample $\mathbf{x}_i^{m_1}$, we first obtain the modality-specific $\mathbf{f}_i^{m_1}$ with the 2D ResNet. Then, we compute the similarity between $\mathbf{f}_i^{m_1}$ and each prototype $\mathbf{P}_{t,c_t}^{m_1}$ in $\mathbf{P}^{m_1}$. The similarity is measured using the cosine similarity:

$$s(\mathbf{f}_i^{m_1}, \mathbf{P}_{t,c_t}^{m_1}) = \frac{\mathbf{f}_i^{m_1} \cdot \mathbf{P}_{t,c_t}^{m_1}}{||\mathbf{f}_i^{m_1}|| \cdot ||\mathbf{P}_{t,c_t}^{m_2}||} . \tag{3}$$

**Feature synthesis.** We then synthesize the features of missing modality data $\tilde{\mathbf{f}}^{m_2}$ with $P^{m_2}$ using the previously computed similarity weights. In addition, modality-specific, disease-wise learnable weights $\pi_t^{m_2}$ are applied to account for the contribution of each modality to the disease $t$ to get the synthetic features for $T$ diseases via:

$$\tilde{\mathbf{f}}_i^{m_2} = \sum_{t=1}^T \sum_{c_t=1}^{C_t} \pi_t^{m_2} \cdot s(\mathbf{f}_i^{m_1}, \mathbf{P}_{t,c_t}^{m_1}) \cdot \mathbf{P}_{t,c_t}^{m_2} . \tag{4}$$

### A.2.4 PERSONALIZED MULTIMODAL FEATURE FUSION.

Intuitively, the modality-specific features extracted directly from the input are more reliable than those generated via the proposed personalized feature completion with the prototypes. Thus, it is desired to fuse modality-specific features from samples with various modality combinations in different ways. To this end, we propose personalized multimodal feature fusion. In detail, we denote the modality-specific features of sample $i$ to be fused as:

$$\begin{aligned} \bar{\mathbf{f}}_i^{m_1} &= \delta_i^{m_1} \cdot \mathbf{f}_i^{m_1} + (1 - \delta_i^{m_1}) \cdot \tilde{\mathbf{f}}_i^{m_1} , \\ \bar{\mathbf{f}}_i^{m_2} &= \delta_i^{m_2} \cdot \mathbf{f}_i^{m_2} + (1 - \delta_i^{m_2}) \cdot \tilde{\mathbf{f}}_i^{m_2} . \end{aligned} \tag{5}$$

Inspired by SKNet (Li et al., 2019), we fuse the two modality-specific feature and obtain a compact multimodal feature descriptor $z$ via:

$$\mathbf{z}_i = RELU\left(BN\left(FC_{[\delta_i^{m_1}, \delta_i^{m_2}]}\left(\bar{\mathbf{f}}_i^{m_1} + \bar{\mathbf{f}}_i^{m_2}\right)\right)\right) , \tag{6}$$

where $[\delta_i^{m_1}, \delta_i^{m_2}]$ indicates the combination of modalities and $FC_{[\delta_i^{m_1}, \delta_i^{m_2}]}$ represents the full connection layer which is modality combination aware. In other words, samples with the same modality combination share the same full connection layer while samples with different modality combinations employ different full connection layers. Then, we learn the channel-wise importance weights for modality-specific features with the guidance of $\mathbf{z}_i$ via:

$$
\begin{aligned}
\mathbf{w}_i^{m_1} &= \frac{exp\left(FC_{[\delta_i^{m_1}]}(\mathbf{z}_i)\right)}{exp\left(FC_{[\delta_i^{m_1}]}(\mathbf{z}_i)\right) + exp\left(FC_{[\delta_i^{m_2}]}(\mathbf{z}_i)\right)}, \\
\mathbf{w}_i^{m_2} &= \frac{exp\left(FC_{[\delta_i^{m_2}]}(\mathbf{z}_i)\right)}{exp\left(FC_{[\delta_i^{m_1}]}(\mathbf{z}_i)\right) + exp\left(FC_{[\delta_i^{m_2}]}(\mathbf{z}_i)\right)},
\end{aligned}
\tag{7}
$$

where $FC_{[\delta_i^{m_1}]}$ and $FC_{[\delta_i^{m_2}]}$ represent the full connection layer applied to $\mathbf{z}_i$ to estimate the importance for each modality respectively. Finally, we fuse the modality-specific features according to their importance weights and obtain the fused multimodal features $\mathbf{f}_i^{mm}$ via:

$$
\mathbf{f}_i^{mm} = \mathbf{w}_i^{m_1} \odot \bar{\mathbf{f}}_i^{m_1} + \mathbf{w}_i^{m_2} \odot \bar{\mathbf{f}}_i^{m_2} \ ,
\tag{8}
$$

where $\odot$ is Hadamard product. With fused features $\mathbf{f}_i^{mm}$, we employ a simple multi-label classifier $cls$ to predict the risks of retinal diseases via:

$$
\hat{\mathbf{y}}_i = cls(\mathbf{f}_i^{mm}).
\tag{9}
$$

### A.3 DETECTION PERFORMANCES FOR EACH DISEASE ON $\mathcal{D}_A$

In this subsection, we provide the class-wise performances on $\mathcal{D}_A$ in Table 11. We can observe that in terms of the balanced evaluation metrics $F1$ and $Kappa$, our proposed baseline PersonNet achieves best consistently on the three diseases. Although the number of positive samples for glaucoma in $\mathcal{D}_A$ is higher than that for macular edema (ME), the $Kappa$ score for glaucoma detection is significantly lower (33.08%) compared to that for ME detection (48.42%). This indicates that glaucoma detection is more challenging than ME detection, which aligns with observations from clinical practice.

| Methods | Acc | | | | F1 | | | | Kappa | | | |
|---|---|---|---|---|---|---|---|---|---|---|---|---|
| | ME | DR | GL | mAcc | ME | DR | GL | mF1 | ME | DR | GL | mKappa |
| MultiModel | 95.49±0.22 | 94.31±0.39 | 90.30±0.46 | 93.37±0.13 | 43.88±4.18 | 65.89±1.94 | 46.61±1.42 | 52.13±0.97 | 41.54±4.28 | 62.79±2.14 | 25.74±3.14 | 43.36±1.23 |
| Concat | 95.53±0.26 | 94.52±0.34 | 87.99±0.56 | 92.68±0.33 | 48.34±1.62 | 67.21±1.30 | 48.46±0.86 | 54.67±0.85 | 46.01±1.71 | 64.22±1.47 | 29.03±0.91 | 46.42±1.00 |
| Sum | 95.32±0.28 | 94.15±0.39 | 87.69±1.36 | 92.39±0.45 | 45.04±3.60 | 65.63±1.66 | 48.14±2.40 | 52.94±0.46 | 42.60±3.74 | 62.43±1.87 | 30.17±3.54 | 45.07±1.27 |
| LF | 95.78±0.46 | 94.35±0.57 | 87.31±0.59 | 92.48±0.21 | 49.97±2.72 | 66.26±2.58 | 48.99±0.99 | 55.07±0.84 | 47.78±2.92 | 63.18±2.88 | 30.21±2.84 | 47.06±1.05 |
| FiLM | 95.57±0.25 | 94.32±0.56 | 89.33±0.54 | 93.07±0.40 | 47.80±1.79 | 66.51±2.52 | 50.33±1.91 | 54.88±1.05 | 45.49±1.86 | 63.41±2.79 | 31.92±1.68 | 46.94±1.32 |
| BiGated | 95.47±0.23 | 94.52±0.26 | 87.74±0.89 | 92.58±0.33 | 45.39±3.73 | 67.81±0.99 | 49.67±1.48 | 54.29±1.15 | 43.03±3.78 | 64.82±1.13 | 30.43±1.57 | 46.09±1.40 |
| MMCNN | 95.57±0.32 | 94.44±0.22 | 88.86±1.30 | 92.96±0.42 | 46.24±3.97 | 65.73±1.76 | 49.57±2.77 | 53.84±1.09 | 43.93±4.08 | 62.71±1.86 | 31.35±3.70 | 46.00±1.17 |
| LFM | 95.86±0.30 | 94.21±0.34 | 89.71±0.41 | 93.26±0.12 | 42.34±5.37 | 63.59±1.67 | 47.04±2.20 | 50.99±2.14 | 40.25±5.48 | 60.46±1.84 | 25.81±2.04 | 42.17±2.12 |
| PersonNet | 95.50±0.66 | 94.88±0.24 | 89.77±0.54 | 93.38±0.30 | 50.74±2.01 | 68.92±1.21 | 51.42±1.68 | 57.03±0.96 | 48.42±2.25 | 66.13±1.31 | 33.08±2.46 | 49.21±1.23 |
| LCR | 95.58±0.40 | 94.60±0.34 | 89.43±0.51 | 93.20±0.18 | 46.69±2.56 | 67.20±2.51 | 50.41±2.21 | 54.76±0.71 | 44.39±2.75 | 64.27±2.68 | 30.50±3.55 | 46.39±0.90 |
| MMANet | 95.30±0.35 | 94.23±0.39 | 88.01±0.52 | 92.51±0.22 | 46.89±2.16 | 66.50±1.99 | 49.02±1.96 | 54.13±1.13 | 44.44±2.33 | 63.34±2.20 | 29.15±2.06 | 45.64±1.26 |
| ShaSpec | 94.81±0.45 | 92.74±0.40 | 85.35±1.11 | 90.97±0.42 | 42.59±2.37 | 60.47±1.07 | 46.92±1.82 | 49.99±1.26 | 39.91±2.43 | 56.53±1.20 | 26.14±2.81 | 40.83±1.57 |
| MLA | 95.83±0.33 | 94.45±0.21 | 89.21±0.47 | 93.16±0.30 | 47.70±3.42 | 67.67±1.20 | 50.33±0.70 | 55.23±1.44 | 45.54±3.48 | 64.64±1.30 | 32.29±2.64 | 47.49±1.43 |
| DMRNet | 93.91±0.86 | 92.55±0.89 | 84.46±2.78 | 90.31±1.38 | 44.12±1.22 | 60.88±1.51 | 48.13±1.74 | 51.04±1.05 | 41.17±1.32 | 56.95±1.84 | 27.31±1.77 | 41.81±0.82 |
| PersonNet | 95.50±0.66 | 94.88±0.24 | 89.77±0.54 | 93.38±0.30 | 50.74±2.01 | 68.92±1.21 | 51.42±1.68 | 57.03±0.96 | 48.42±2.25 | 66.13±1.31 | 33.08±2.46 | 49.21±1.23 |

Table 11: Performances for each disease of our PersonNet and the compared 13 baselines on the test set of $\mathcal{D}_A$. "ME", "DR", and "GL" represent macular edema, diabetic retinopathy and glaucoma respectively. Means and standard deviations over five trials are reported.

### A.4 ABLATION STUDIES

**Ablation Study on Different Design Choices**. We first conducted ablation studies on the validation set of $\mathcal{D}_A$ to investigate how the proposed personalized missing modality completion module and personalized multimodal feature fusion module contribute to the performance. The results are reported in the table below. The performances on all samples in the validation set are shown in Tab. 12. We observe that: (1) with the proposed completion module, $mAcc$, $mF1$ and $mKappa$ consistently increase compared to the baseline; (2) the proposed fusion module also leads to performance gains; (3) combined together, our PersonNet substantially outperforms the baseline.

| Method | mAcc (%) | mF1 (%) | mKappa (%) |
|---|---|---|---|
| Baseline | $93.19 \pm 0.40$ | $53.47 \pm 2.10$ | $44.96 \pm 2.84$ |
| + Personalized Missing Modality Completion | $93.47 \pm 0.19$ | $54.70 \pm 1.68$ | $46.79 \pm 2.05$ |
| + Personalized Multimodal Fusion | $93.40 \pm 0.14$ | $53.75 \pm 0.60$ | $46.59 \pm 0.77$ |
| + Both (PersonNet) | $93.93 \pm 0.12$ | $56.27 \pm 1.75$ | $49.32 \pm 2.02$ |

Table 12: Performances on all samples in the validation set of $\mathcal{D}_A$.

| Method | mAcc (%) | mF1 (%) | mKappa (%) |
|---|---|---|---|
| Baseline | $93.19 \pm 0.40$ | $53.47 \pm 2.10$ | $44.96 \pm 2.84$ |
| + Personalized Missing Modality Completion | $94.63 \pm 0.30$ | $49.37 \pm 2.70$ | $44.15 \pm 2.38$ |
| + Personalized Multimodal Fusion | $93.40 \pm 0.14$ | $53.75 \pm 0.60$ | $46.59 \pm 0.77$ |
| + Both (PersonNet) | $93.93 \pm 0.12$ | $56.27 \pm 1.75$ | $49.32 \pm 2.02$ |

Table 13: Performances on samples with only UWF-SLO in the validation set of $\mathcal{D}_A$.

| Method | mAcc (%) | mF1 (%) | mKappa (%) |
|---|---|---|---|
| Baseline | $89.73 \pm 0.50$ | $53.89 \pm 0.79$ | $44.06 \pm 0.67$ |
| + Personalized Missing Modality Completion | $90.69 \pm 0.44$ | $54.31 \pm 1.91$ | $44.69 \pm 2.68$ |
| + Personalized Multimodal Fusion | $90.51 \pm 0.37$ | $53.65 \pm 1.01$ | $45.08 \pm 1.23$ |
| + Both (PersonNet) | $91.21 \pm 0.41$ | $53.74 \pm 1.97$ | $45.18 \pm 2.38$ |

Table 14: Performances on samples with only Macular OCT in the validation set of $\mathcal{D}_A$.

| Method | mAcc (%) | mF1 (%) | mKappa (%) |
|---|---|---|---|
| Baseline | $93.09 \pm 0.68$ | $51.41 \pm 5.61$ | $37.92 \pm 4.19$ |
| + Personalized Missing Modality Completion | $93.22 \pm 0.37$ | $51.93 \pm 3.30$ | $40.33 \pm 2.66$ |
| + Personalized Multimodal Fusion | $93.08 \pm 0.28$ | $49.48 \pm 1.39$ | $38.14 \pm 3.00$ |
| + Both (PersonNet) | $93.49 \pm 0.40$ | $51.67 \pm 1.56$ | $39.36 \pm 1.45$ |

Table 15: Performances on samples with UWF-SLO and Macular OCT in the validation set of $\mathcal{D}_A$.

To investigate how the two proposed modules contribute to performance improvements, we report the results on samples with only UWF-SLO images, samples with only macular OCT, and samples with both modalities in the validation set of $\mathcal{D}_A$, as shown in Tab. 13, Tab. 14, and Tab. 15, respectively. From these results, we observe that:

- Compared to the baseline, the Personalized Missing Modality Completion module improves performance on samples with macular OCT and those with both modalities across all three metrics, but slightly degrades $mF1$ and $mKappa$ on samples with only UWF-SLO;

- Compared to the baseline, the Personalized Multimodal Fusion module improves performance on samples with only UWF-SLO, achieves comparable performance on samples with only macular OCT, but slightly degrades performance on samples with both modalities;

- With both modules, our PersonNet outperforms the baseline in most cases, except for $mF1$ on samples with only macular OCT.

## A.5 WHY INCOMPLETE MULTIMODAL LEARNING?

To demonstrate that missing multimodal learning provides meaningful advantages over single-modal and complete multimodal learning approaches, we conduct fundamental premise validation. As shown in Tab. 3, in the training set of $\mathcal{D}_A$, there are 1,722 samples with only UWF-SLO images, 1,216 samples with macular OCT images, and 808 samples with two modalities to separately train three models. Thus, we can use the UWF-SLO images from 2,530 samples (1,722 + 808), macular OCT images from 2,024 samples (1,216 + 808), and paired UWK-SLO and macular OCT images from 808 samples to seperately train two single-modality model and one complete-modality model for retinal disease detection. We denote them as UWF-SLO-model, Macular-OCT-model, UWF-SLO & OCT-model and compare the performances in Tab. 16. As shown, PersonNet consistently

outperforms the two single-modality and complete-modality models across all subsets, demonstrating the effectiveness of our personalized missing modality learning approach. This also indicates that leveraging both modality-complete and modality-incomplete samples during training contributes to improved performances.

| Sample Type | PersonNet | UWF-SLO-model | Macular OCT-model | UWF-SLO & OCT-model |
|---|---|---|---|---|
| Only UWF-SLO | $49.45 \pm 2.16$ | $46.75 \pm 2.16$ | N.A. | N.A. |
| Only Macular OCT | $43.40 \pm 2.15$ | N.A. | $37.92 \pm 4.14$ | N.A. |
| UWF-SLO & OCT | $48.36 \pm 2.57$ | N.A. | N.A. | $37.64 \pm 2.27$ |

Table 16: Comparison of $mKappa$ for PersonNet and Single-/Multimodal Models on $\mathcal{D}_A$.

Similarly, as shown in Tab. 3, in the training set of dataset $\mathcal{D}_B$, there are 1,830 eyes with only UWF-SLO images, 1,344 eyes with only disc OCT images and 700 eyes with both modalities. Thus, we separately train two single-modality models using the UWF-SLO images from 2530 eyes (1,830+700) and the disc OCT images from 2,044 eyes (1,344+700), and train a complete-modality model using the paired UWF-SLO and disc OCT images from 700 eyes. We denote these three models as UWF-SLO-model, Disc-OCT-model and UWF-SLO & OCT-model respectively. The performances are reported in Tab. 17, which again confirm the superior performance of PersonNet across most settings.

| Sample Type | PersonNet | UWF-SLO-model | Disc OCT-model | UWF-SLO & OCT-model |
|---|---|---|---|---|
| Only UWF-SLO | $29.09 \pm 6.09$ | $31.19 \pm 3.23$ | N.A. | N.A. |
| Only Disc OCT | $56.54 \pm 2.62$ | N.A. | $55.05 \pm 1.73$ | N.A. |
| UWF-SLO & OCT | $50.66 \pm 3.82$ | N.A. | N.A. | $38.28 \pm 3.59$ |

Table 17: Comparison of $Kappa$ for PersonNet and Single-/Multimodal Models on $\mathcal{D}_B$.

### A.6 Is PersonNet Fair to Different Demographic Subgroups?

To explore whether PersonNet is fair to different demographic subgroups, we categorize samples into three subgroups regarding their age: (0, 40], (40, 70], and over 70. Tab. 18 reports performances of PersonNet and the baseline on $\mathcal{D}_B$. We observe that there is a noticeable performance gap regarding $F1$ and $Kappa$ across the three subgroups. Therefore, developing fair multimodal learning methods is critical to improve fairness across diverse age subgroups.

| Age Group | Acc | | F1 | | Kappa | |
|---|---|---|---|---|---|---|
| | PersonNet | Baseline-Concat | PersonNet | Baseline-Concat | PersonNet | Baseline-Concat |
| (0, 40] | $89.72 \pm 0.73$ | $89.04 \pm 0.94$ | $62.96 \pm 2.47$ | $60.14 \pm 2.15$ | $46.39 \pm 2.55$ | $43.64 \pm 2.71$ |
| (40, 70] | $90.05 \pm 0.69$ | $89.39 \pm 0.64$ | $65.48 \pm 2.67$ | $62.51 \pm 2.41$ | $63.22 \pm 3.43$ | $60.50 \pm 2.40$ |
| > 70 | $79.33 \pm 1.90$ | $77.65 \pm 1.42$ | $48.85 \pm 1.60$ | $47.68 \pm 1.16$ | $47.16 \pm 4.77$ | $43.49 \pm 3.16$ |

Table 18: Performances of different age subgroups for PersonNet and Baseline on $\mathcal{D}_B$.

Similarly, we report performances on each gender group in Tab. 19. Although the gender distribution in M$^3$Ret is nearly balanced, a noticeable performance gap regarding $F1$ and $Kappa$ remains. This again highlights the need for fair multimodal learning.

| Gender | Acc | | F1 | | Kappa | |
|---|---|---|---|---|---|---|
| | PersonNet | Baseline-Concat | PersonNet | Baseline-Concat | PersonNet | Baseline-Concat |
| Female | $88.41 \pm 0.92$ | $87.90 \pm 0.90$ | $62.27 \pm 1.46$ | $59.63 \pm 1.95$ | $49.70 \pm 4.19$ | $48.03 \pm 1.90$ |
| Male | $89.80 \pm 0.18$ | $88.82 \pm 0.90$ | $68.26 \pm 1.00$ | $64.86 \pm 2.23$ | $64.55 \pm 2.63$ | $61.05 \pm 3.63$ |

Table 19: Performances of different gender subgroups for PersonNet and Baseline on $\mathcal{D}_B$.

