# OpenReview forum: "M$^3$Ret: A Mixed Multimodal Image Dataset and Benchmark for Personalized Multi-Retinal Disease Detection"
_ICLR.cc/2026/Conference — Submitted to ICLR 2026_

### Official Review · Reviewer_bWtN · 2025-10-29

**Soundness:** 3
**Presentation:** 3
**Contribution:** 3
**Rating:** 6
**Confidence:** 3

**Summary:**

This paper addresses the gap between existing multimodal ophthalmic datasets/models and real-world clinical needs, where personalized examinations lead to diverse modality combinations and coexisting retinal diseases. Although this article still has some weaknesses in terms of the extensiveness and quality of the dataset, it has made significant progress compared to past work. In my view, this paper is marginally above the acceptance bar of ICLR.

**Strengths:**

1. Unlike existing datasets (e.g., FairVision, MMC-AMD) that only support single-disease detection or fixed modality combinations, M³Ret includes 7 mixed modality combinations (unimodal, bimodal, trimodal) and multi-label disease labels, fully reflecting personalized clinical examinations.

2. With 8,558 eye samples, it is one of the largest multimodal ophthalmic datasets. Its disease prevalence rates are consistent with real-world statistics (e.g., close to Teo et al.’s 4.07% ME and 6.17% DR), avoiding bias from overrepresented diseases (e.g., FairVision’s 48.7% glaucoma).

3. Comprehensive experiments cover 13 baseline methods across complete and incomplete multimodal learning and explore subgroup analysis (age/gender fairness), and single-vs-multimodal comparisons, revealing limitations of existing SOTA and providing insights into model behavior.

**Weaknesses:**

1. M³Ret only includes three diseases (DR, ME, glaucoma), excluding other common retinal diseases (e.g., age-related macular degeneration (AMD)), reducing its applicability to broader clinical scenarios.

2. M³Ret is collected from a single hospital, lacking diversity in patient demographics (e.g., ethnicity, regional medical practices) and imaging devices (only Optos Panoramic 200 for UWF-SLO, CIRRUS HD-OCT 500 for OCT), limiting generalization to other clinical settings.

3. A portion of labels are derived from treatment records (not direct diagnoses) or marked as “unclear”, which may introduce noise into training.

**Questions:**

1. PersonNet’s fusion module uses modality combination-aware fully connected layers. How many such layers are there (one per combination?), and how do you avoid overfitting given the varying sample sizes across combinations (e.g., Tri-modal only has 118 samples)?

2. The paper suggests future work report computational metrics (e.g., FLOPS, FPS). Have you tested PersonNet’s performance on edge devices (e.g., mobile GPUs) to evaluate its deployment potential in primary care settings?

---

> ### Author Response · Authors · 2025-11-13
> **Reply to Reviewer bWtN**
>
> We thank reviewers for their timely feedback and valuable comments. Below is our point-by-point response.
>
> **Weakness1 & 2:** M$^3$Ret only includes three diseases and was collected from a single hospital, reducing its applicability to broader clinical scenarios and limiting generalization to other clinical settings.
>
> **Reply:** Thanks for pointing this out. Compared to existing publicly available multimodal ophthalmic datasets, which typically focus on a single specific retinal disease, M³Ret covers three retinal diseases and two modalities by two different imaging devices, providing a more comprehensive resource for research. In the future, we plan to collaborate with ophthalmologists from other countries to collect additional data encompassing a wider range of retinal diseases and more diverse ethnicities. Nevertheless, we still think it is important and timely to release the data collected from real clinical workflows and benchmarks to foster research in this field.
>
> **Weakness3:**  A portion of labels are derived from treatment records (not direct diagnoses) or marked as “unclear”, which may introduce noise into training.
>
> **Reply:**  Thank you for your valuable comment. In the health domain, labels must be highly reliable. To ensure this, our dataset was collected from the ophthalmic outpatient department of the hospital, and over 90% of the samples have clear diagnostic decisions recorded in the electronic medical record system, complemented by treatment records when the diagnoses were positive. These diagnostic decisions were made by experienced ophthalmologists based on a combination of clinical examinations, including visual field tests, intraocular pressure measurements, slit-lamp examinations, SLO imaging, OCT imaging, and others.
> For samples without clear diagnostic labels, if two experienced ophthalmologists could confidently determine the disease labels based on detailed treatment and medication records, their labels were assigned accordingly. Otherwise, the disease labels were marked as “unclear.” As a result, only approximately 5.2% of the samples have unclear labels for partial disease classes, demonstrating that the labeling process was conducted carefully to ensure high reliability in our dataset.
> Furthermore, for samples with an “unclear” label for one disease but “clear” labels for other diseases, we still use the “clear” labels in model training. Specifically, the loss is calculated only over disease classes with clear labels, while disease classes with “unclear” labels are excluded from loss computation. Therefore, the presence of “unclear” labels does not introduce noise into model training.
>
> **Question 1:**  PersonNet’s fusion module uses modality combination-aware fully connected layers. How many such layers are there (one per combination?), and how do you avoid overfitting given the varying sample sizes across combinations (e.g., Tri-modal only has 118 samples)?
>
> **Reply:**  Thank you very much for these questions. In clinical practice, UWF-SLO images, macular OCT, and their paired combinations are commonly used to diagnose macular edema (ME), diabetic retinopathy (DR), and glaucoma. UWF-SLO images, disc OCT images, and their paired combinations are primarily used for diagnosing glaucoma because glaucoma results in degradation of optic nerves which gathering in optic disc while ME and DR do not. This is the reason why only a very limited number of patients undergo all three examinations (1.4%) or both macular and disc OCT scans (0.7%) as shown in Table 2. Accordingly, we reorganise the data to two sub-datasets: $\mathcal{D}_A$ and $\mathcal{D}_B$. $\mathcal{D}_A$ consists of three combinations of UWF-SLO image and macular OCT images and is used for the detection of ME, DR and glaucoma and $\mathcal{D}_B$ consists of three combinations of UWF-SLO images and disc OCT images and is used for the detection of glaucoma. The summary can be found in Table 3.  Since each sub-dataset only contains three combinations, we naturally set the number of MLP in Figure 5 to three. For samples in Tri-modal, the UWF-SLO and macular OCT are included in $\mathcal{D}_A$ while UWF-SLO and disc OCT are included in  $\mathcal{D}_B$.
>
> **Question 2:** Have you tested PersonNet’s performance on edge devices (e.g., mobile GPUs) to evaluate its deployment potential in primary care settings?
>
> **Reply:** Thank you very much for the question. We have not tested our model’s performance on edge devices. Probably in the health domain, the better option for deploying the disease classification model is a client–server architecture rather than edge computing architecture.

---

### Official Review · Reviewer_Zydr · 2025-10-29

**Soundness:** 2
**Presentation:** 2
**Contribution:** 2
**Rating:** 4
**Confidence:** 5

**Summary:**

The paper introduces M3Ret, a mixed-modality ophthalmic dataset with multiple modality combinations.

It also proposes PersonNet, a baseline for personalized multi-disease detection with a memory-bank completion and fusion strategy.

Tens of multimodal methods are benchmarked on this dataset.

**Strengths:**

+ The problem of multiple retinal disease and AI diagnosis is timely.

+ The dataset scale is attractive.

+ Overall the paper is easy-to-follow and clearly-presented.

**Weaknesses:**

- This paper claims “first” to support diverse combinations for eye disease, which is not true. Some prior works are listed as follows.

[1] Harvard Glaucoma Detection and Progression: A Multimodal Multitask Dataset and Generalization-Reinforced Semi-Supervised Learning. [https://arxiv.org/abs/2308.13411]

[2] FairVision: Equitable Deep Learning for Eye Disease
Screening via Fair Identity Scaling [https://github.com/Harvard-Ophthalmology-AI-Lab/FairVision]

- The center and vendor type in this work is not clearly detailed, and may be rather limited. Instead, cross-site testing and generalization to other vendors are needed, to enrich the diversity of the benchmark.

- For the evaluation metrics, it may be more rationale to consider precision, recall, F1-score and their class-wise metric.

- Besides, please do more analysis and show the baseline outcomes on the per-class per-disease performance.

- The technique novelties of the proposed method are limited. Specifically, memory-bank completion with class-wise prototypes is a straightforward instantiation of missing-modality completion.

- Since this paper considers the imcomplete modality settings, some stronger generative baselines should also be compared.

- The experiments and validation seem insufficient. For example, the sensitivity to prototype bank size/update. Besides, how does the class imbalance problem impact the performance?

**Questions:**

Please refer to the weakness section, and address these concerns point-by-point.

---

> ### Author Response · Authors · 2025-11-13
>
> We thank reviewers for their timely feedback and valuable comments. Below is our point-by-point response.
>
> **Weakness1:**  This paper claims “first” to support diverse combinations for eye disease, which is not true. Prior works: FairVision, Harvard-GDP
>
> **Reply:** We have noticed the datasets FairVision and list its details in Table1. We would like to illustrate the differences between FairVision and our M$^3$Ret from two aspects. **First**, FairVision contains three separate sub-datasets, each collected for detecting a specific disease. This indicates that FairVision is, in essence, a single-label disease classification dataset. Differently, our M$^3$Ret is a multi-label disease classification dataset. **Second**, in FairVision, each sub-dataset includes paired SLO and OCT images cantered on the same local region usually size of 6mm$\times$6mm: either the macula or the optic disc. These two modalities are captured during the same scanning session using the same device. This means that, in clinical practice, the SLO and OCT images in FairVision always appear as pairs. Of course, we can artificially create diverse combinations for research purposes, but the necessity of doing so remains uncertain.  Differently, in our M$^3$Ret, the two modalities, i.e., UWF-SLO and OCT images are captured by two different devices. UWF-SLO images cover an ultra-wide field of view of the retina and OCT images provide cross-sectional information over a local region size of 6mm$\times$6mm. Personalized clinical examinations result in diverse combinations of UWF-SLO and OCT images, which are critical for personalized diagnosis and treatment, and are therefore more common in real-world clinical practice.
>
> Harvard-GDP was collected only for glaucoma detection and progression. It consists of two modalities: VF tests (a non-imaging modality) and RNFLT derived from 3D OCT images. These two modalities are highly specific to glaucoma, particularly for monitoring its progression. In contrast, our M$^3$Ret dataset is designed for multi-label disease classification and covers three different retinal diseases.
>
> Overall, we can conclude that our M$^3$Ret is the first dataset that considers diverse modality combinations and supports personalized multi-retinal disease detection.
>
> **Weakness2:**  The center and vendor type in this work is not clearly detailed and may be rather limited. Instead, cross-site testing and generalization to other vendors are needed, to enrich the diversity of the benchmark.
>
> **Reply:** Due to the double-blind peer review policy, the center where the study was conducted was not disclosed and it will be disclosed after the paper is accepted. The vendor types can be found in lines 188-190. UWF-SLO images were captured using the Optos Panoramic 200 scanning laser ophthalmoscope. OCT images were acquired using the CIRRUS HD-OCT 500 device. As M$^3$Ret is currently the only one multimodal dataset with UWF-SLO and OCT images, it is impossible to conduct cross-site testing. In future, we will collaborate with ophthalmologists from other countries to collect additional data captured by different vendors and enrich the diversity of the benchmark.
>
> **Weakness3:**  For the evaluation metrics, it may be more rationale to consider precision, recall, F1-score and their class-wise metric.
>
> **Reply:** The F1-score and class-wise metric have been reported in Table 11. As F1-score is a harmonic mean of precision and recall, we do not report the recall and precision.
>
> **Weakness4:**  Besides, please do more analysis and show the baseline outcomes on the per-class per-disease performance.
>
> **Reply:** The per-class performances of baselines have been reported in Table 11.
>
> **Weakness5:**  The technique novelties of the proposed method are limited. Specifically, memory-bank completion with class-wise prototypes is a straightforward instantiation of missing-modality completion.
>
> **Reply:** This submission was made to the Dataset and Benchmark track, with contributions threefold as outlined in the Introduction: (1) the novel dataset M$^3$Ret, (2) the strong baseline method PersonNet, and (3) the benchmark itself. While the proposed baseline PersonNet is simple and intuitive, it achieves superior performance. Its effectiveness has been demonstrated in the Experiments section.

---

> > ### Author Response · Authors · 2025-11-14
> >
> > **Weakness6:**  Since this paper considers the incomplete modality settings, some stronger generative baselines should also be compared.
> >
> > **Reply:** The lesions/abnormalities such as microaneurysm and neovascularization etc. associated to retinal diseases are usually extremely tiny and subtle. Generative methods, such as diffusion models, naturally have the ability to ‘purify’ images [1], often mistakenly treating tiny and subtle lesions or abnormalities as noise. As a result, they may inadvertently remove these tiny but clinically important lesions/abnormalities. Consequently, these clinically important lesions/abnormities may be inadvertently removed. Synthesizing missing modalities that preserve such tiny and subtle lesions using diffusion models remains a significant challenge.
> >
> > [1]: Weili Nie et al. Diffusion models for adversarial purification. ICML, 2022.
> >
> > **Weakness7:**  The experiments and validation seem insufficient. For example, the sensitivity to prototype bank size/update. Besides, how does the class imbalance problem impact the performance?
> >
> > **Reply:** Thank you for point this out. We use EMA to update the prototypes, thus the memory bank can be set to 1. We compare three different update strategies:
> >
> > Update1: update when each sample comes
> >
> > Update2: update when each batch samples come (the one we adopted in our paper)
> >
> > Update3: update after each epoch
> >
> > Below are the performances on  for the three update strategies:
> >
> > | Update| mKappa        | mF            | mAcc         |
> > |----------|---------------|---------------|--------------|
> > | Update1  | 46.53±1.26    | 54.61±1.41    | 93.51±0.22   |
> > | Update2 (ours)  | 46.79±2.05    | 54.70±1.68    | 93.47±0.19   |
> > | Update3  | 46.26±0.99    | 54.18±1.16    | 93.43±0.25   |
> > From the above table, we can see that the effects of different update strategies are not significant.

---

### Official Review · Reviewer_nhTT · 2025-10-30

**Soundness:** 2
**Presentation:** 2
**Contribution:** 2
**Rating:** 4
**Confidence:** 3

**Summary:**

This paper addresses the challenges of incomplete modality combinations and multi-disease detection in ophthalmic multimodal image analysis by proposing the $M^3Ret$ dataset and the $PersonNet$ method. $M^3Ret$ comprises retinal images with diverse modality combinations and supports the detection of three retinal diseases. $PersonNet$ handles varying modality combinations through missing modality completion and personalized fusion strategies. The study conducts a benchmark evaluation of the proposed method alongside existing multimodal learning approaches on the $M^3Ret$ dataset and provides an analysis of the results.

**Strengths:**

1. The $M^3Ret$ dataset holds certain advantages in scale and encompasses diverse imaging modality combinations, which better reflects real-world clinical heterogeneity compared to existing datasets that only include complete modality pairs.

2. This paper identifies a limitation in current ophthalmic multimodal research, specifically, its reliance on fixed, complete modality combinations. And introduces the problem of personalized detection, which is more aligned with practical clinical scenarios.

**Weaknesses:**

1. All evaluations of the proposed PersonNet method were conducted solely on the $M^3Ret$ dataset, with no testing performed on external datasets. This limitation fails to demonstrate the method's generalization capability and broader effectiveness.

2. The core components of PersonNet, the class-wise prototype-based missing modality completion and the SKNet-inspired feature fusion—represent relatively straightforward and conventional approaches within the existing literature on incomplete multimodal learning. The paper does not sufficiently justify the significant innovation of the proposed method compared to existing techniques.

3. Critical ablation studies are reported only on the validation set, lacking final verification on the test set. This omission undermines the reliability of the conclusions drawn.

**Questions:**

As indicated in the weaknesses.

---

> ### Author Response · Authors · 2025-11-13
> **Ablation studies can only be conducted on validation set**
>
> We thank reviewers for their timely feedback and valuable comments. Below is our point-by-point response.
>
> **W1:** All evaluations of the proposed PersonNet method were conducted solely on the  dataset, with no testing performed on external datasets. This limitation fails to demonstrate the method's generalization capability and broader effectiveness.
>
> **Reply:**  To the best of our knowledge, M$^3$Ret is the first multimodal dataset containing high-resolution UWF-SLO and OCT images. Therefore, it is currently not feasible to perform cross-dataset validation to demonstrate generalizability across different imaging protocols and equipment.
>
> **W2:** The core components of PersonNet, the class-wise prototype-based missing modality completion and the SKNet-inspired feature fusion—represent relatively straightforward and conventional approaches within the existing literature on incomplete multimodal learning. The paper does not sufficiently justify the significant innovation of the proposed method compared to existing techniques.
>
> **Reply:**  This submission was made to the Dataset and Benchmark track, with contributions threefold as outlined in the Introduction: (1) the novel dataset M$^3$Ret, (2) the strong baseline method PersonNet, and (3) the benchmark itself. While the proposed baseline PersonNet is simple and intuitive, it achieves superior performance. Its effectiveness has been demonstrated in the Experiments section.
>
> **W3:** Critical ablation studies are reported only on the validation set, lacking final verification on the test set. This omission undermines the reliability of the conclusions drawn.
>
> **Reply:**  Ablation studies must be conducted on the validation set to avoid test set contamination and the test set should be reserved solely for final evaluation of model performance. This principle has been illustrated in many foundmental books about machine learning and pattern recognition, e.g., "Pattern Recognition and Machine Learning" by Christopher M. Bishop which says  "If data is plentiful, then one approach is simply to use some of the available data to train a range of models, or a given model with a range of values for its complexity parameters, and then to compare them on independent data, sometimes called a validation set, and select the one having the best predictive performance. If the model design is iterated many times using a limited size data set, then some over-fitting to the validation data can occur and so it may be necessary to keep aside a third test set on which the performance of the selected model is finally evaluated." We strictly follow this principle and perform our ablation studies on the validation set while comparing our results with state-of-the-art methods on the test set. Previous works, such as the Swin Transformer (ICCV 2021 Best Paper), have also followed this principle.

---

### Official Review · Reviewer_iVNK · 2025-10-31

**Soundness:** 3
**Presentation:** 3
**Contribution:** 3
**Rating:** 4
**Confidence:** 3

**Summary:**

The paper introduces M$^3$Ret, a mixed multimodal ophthalmic dataset of 8,558 eyes / 5,235 patients covering UWF-SLO, macular OCT (128 B-scans), disc OCT (200 B-scans) with seven real-world modality combinations. It defines two benchmark suites (DA: multi-disease ME/DR/Glaucoma; DB: glaucoma) with stratified 6:2:2 patient splits, and provides PersonNet as a baseline supporting incomplete-modality inputs. The benchmark reports comprehensive performance metrics. Authors have release code and a data link and discuss ethics.

**Strengths:**

**S1.** This paper is well-organized and easy to follow.

**S2.** Seven observed modality combinations reflect routine workflows (uni-/bi-/tri-modal sampling), which is rare in this space.

**S3.** The paper reports age/sex distributions and disease prevalence close to epidemiology, supporting external validity. In addition, the provided dataset and code are both complete.

**Weaknesses:**

**W1.** Most labels come from EMR diagnoses; when those are missing, experts infer them from treatment records. Some cases are still marked ‘unclear,’ and there’s no inter-rater protocol, adjudication process, or label-noise audit.

**W2.** DA and DB are reorganizations of the same hospital cohort with stratified 6:2:2 splits, but it is not explicit whether splits are patient-disjoint within and across DA/DB, nor whether tri-modal cases can leak information between tasks.

**W3.** Benchmark scope is misses clinically critical metrics including ROC-AUC, calibration (ECE/Brier), or subgroup performance (e.g., age/sex strata) and analyses.

**W4.** Missing details on OCT resampling/padding and on whether duplicate scans are kept or discarded.

**Questions:**

Please see Weaknesses.

---

> ### Author Response · Authors · 2025-11-14
>
> We greatly appreciate the comments from reviewer iVNK. Please find our feedback below.
>
> **W1.** Most labels come from EMR diagnoses; when those are missing, experts infer them from treatment records. Some cases are still marked ‘unclear,’ and there’s no inter-rater protocol, adjudication process, or label-noise audit.
>
> **Reply:** From Table 9, we can see, for ME and DR, only four and five samples are marked as “unclear” respectively. This means that diagnosis these two diseases are not difficult according to the records in the EMR system. Regarding glaucoma, there are 132 samples marked as “unclear”. The reason is that diagnosing glaucoma, particularly at a very early stage, is highly challenging. This also explains why the diagnostic records of some samples in the EMR system are marked as suspicious, indicating that a definitive diagnosis cannot be made and that follow-up examinations are required. In the health domain, labels must be highly reliable. If two ophthalmologists could not confidently decide the labels for those 132 samples, marking them as “unclear” would be more reliable than labelling them via inter-rater protocol. From this aspect, we can confidently say that the diagnostic labels are highly reliable in our dataset. **Disease diagnosis is highly complex and should be based on comprehensive examinations rather than on inter-rater assessments derived from incomplete records.**
>
> **W2.** DA and DB are reorganizations of the same hospital cohort with stratified 6:2:2 splits, but it is not explicit whether splits are patient-disjoint within and across DA/DB, nor whether tri-modal cases can leak information between tasks.
>
> **Reply:** We take Uni-1 in Table 2 as example and illustrate how to split the samples into traing/val/test sets and ensure no leak information between tasks. For samples in Uni-1, we first selected samples with both DR and ME and randomly split them into train/val/test sets using an approx. rate of 6:2:2. The same procedure was applied to samples with both DR and glaucoma, as well as those with both ME and glaucoma.  Next, we selected samples with only DR and split them in the same way. The same procedure was followed for samples with only ME, glaucoma, or suspicious glaucoma. Finally, samples without any diagnosed disease were also randomly split using the same ratio. In this way, approx. 60% of the positive samples are included in the training set, 20% in the validation set, and 20% in the test set, ensuring balanced representation across subsets.
> For samples in Uni-2, Uni-3 as well as Bi-modal and Tri-modal, we adopt the same strategy to split them into train/val/test. Thereafter, we put the training/val/test samples in Uni-1 and Bi-2, Uni-2 and Bi-3 as well as Bi-1 and Tri-1 to the training/val/test sets of DA. We put the training/val/test samples in Uni-1 and Bi-1, Uni-3 and Bi-3 as well as Bi-2 and Tri-1 to the training/val/test sets of DB. By doing so, training samples in DA do not appear in the validation or test sets of DB, and training samples in DB do not appear in the validation or test sets of DA, thereby preventing any information leakage between DA and DB.

---

> ### Author Response · Authors · 2025-11-14
>
> **W3.** Benchmark scope is misses clinically critical metrics including ROC-AUC, calibration (ECE/Brier), or subgroup performance (e.g., age/sex strata) and analyses.
>
> **Reply:** Thank you for pointing out this. Regarding the subgroup performances, we have reported them in subsection **A.6 IS PersonNet FAIR TO DIFFERENT DEMOGRAPHIC SUBGROUPS?** in Page 17.
>
> ECE and Brier score are not reasonable to evaluate model performance when the class distribution is extremely imbalanced. In such cases, metrics specifically designed for imbalanced data, such as F-score and Kappa, are more appropriate. To illustrate this, consider the following scenario.
>
> Suppose we have 100 samples where 90 are negative samples and 10 are positive samples and we have three models list below:
>
> ModelA predicts 100 samples as negative samples
>
> ModelB predicts 5 negative samples as positive and predict 5 positive samples as negative
>
> ModelC predicts 10 negative samples as positive and correctly predict 10 positive samples as positive
>
> ECE and Brier by ModelA: 0.9 and 0.1
>
> ECE and Brier by ModelB: 0.9 and 0.1
>
> ECE and Brier by ModelC: 0.9 and 0.1
>
> F1-score and Kappa by ModelA: 0, 0
>
> F1-score and Kappa by ModelB: 0.5, 0.444
>
> F1-score and Kappa by ModelC: 0.667, 0.615
>
> From above, it is obvious that ECE and Brier are not reasonable evaluation metrics when the test data is extremely class distribution imbalanced while the evaluation metrics F1 score and Kappa we used in our paper are more reasonable.
>
> Regarding the ROC-AUC, per your suggestion, we report the  ROC-AUC below. We can see that PersonNet outperforms all the compared complete multimodal learning methods and achieves comparable AUC to the second best incomplete multimodal learning method.
>
>
> | Methods                       | ME (%) ± SD | DR (%) ± SD | Glaucoma (%) ± SD | Average (%) ± SD |
> |--------------------------------|------------|------------|-----------------|----------------|
> | **Multimodal**                 | 91.03±1.28 | 92.60±1.33 | 74.71±1.90      | 86.11±0.80     |
> | **Complete multimodal learning** |            |            |                 |                |
> | Concat                         | 91.55±0.97 | 91.85±0.64 | 77.82±0.54      | 87.07±0.30     |
> | Sum                            | 91.56±0.82 | 92.63±0.56 | 78.27±1.12      | 87.49±0.32     |
> | LF                             | 91.93±1.09 | 91.97±0.59 | 78.11±1.86      | 87.34±0.27     |
> | FiLM                           | 92.31±0.58 | 92.96±0.61 | 78.42±0.65      | 87.90±0.41 (second best)    |
> | BiGated                        | 92.46±0.57 | 93.06±0.81 | 77.66±0.81      | 87.73±0.58     |
> | MMCNN                          | 90.63±0.98 | 91.96±0.57 | 77.90±2.07      | 86.83±0.84     |
> | LFM                            | 88.23±3.42 | 90.05±0.55 | 74.98±2.23      | 84.42±1.24     |
> | **PersonNet**                      | 91.72±0.94 | 94.30±2.19 | 78.14±0.87      | 88.06±1.03  (best)   |
> | **Incomplete multimodal learning** |          |            |                 |                |
> | LCR                            | 91.55±0.94 | 92.54±0.39 | 78.32±1.19      | 87.47±0.70     |
> | MMANet                         | 91.45±1.29 | 92.60±0.49 | 77.87±0.34      | 87.31±0.58     |
> | ShaSpec                        | 90.90±1.04 | 91.21±0.41 | 77.47±0.51      | 86.53±0.52     |
> | MLA                            | 92.85±0.65 | 93.19±0.39 | 78.41±0.53      | 88.15±0.29   (best) |
> | DMRNet                         | 87.18±2.46 | 89.97±0.90 | 73.73±1.76      | 83.63±1.03     |
> | **PersonNet**                      | 91.72±0.94 | 94.30±2.19 | 78.14±0.87      | 88.06±1.03   (second best)  |
>
> **W4.** Missing details on OCT resampling/padding and on whether duplicate scans are kept or discarded.
>
> **Reply:** If a sample contains duplicate OCT scans, we retain all scans and treat them as multiple views of the same sample, effectively serving as a form of data augmentation.
>
> We thoroughly reviewed the manuscript and did not find any mention of “resampling.” We would be grateful if you could provide more details about your concerns or the specific context in which resampling might be an issue.
>
> If the “padding” you are concerned with refers to the one we used for data augmentation, we randomly pad zeros in three directions of 3D OCT images and then randomly crop back to the original size to simulate displacement transformations. This makes the trained model more robust to displacement transformations.

---

> > ### Comment · Reviewer_iVNK · 2025-11-26
> >
> > Thank you for the detailed response. Most of my concerns have been addressed.
> >
> > The remaining concerns patient-level deduplication and cross-task alignment. Given that many patients may have multiple visits, repeated OCT scans (or different combinations of modalities), could you clarify whether patient identifiers were used to ensure that all samples from the same patient are strictly contained within a single split and do not appear in both DA and DB?

---

> ### Author Response · Authors · 2025-11-26
> **Reply to Reviewer iVNK**
>
> Many thanks for the discussion and the additional question. In the hospital where we collected our data, each patient in the EMR system has a unique identifier that has a one-to-one correspondence with the patient’s social security ID, but is not the social security ID itself. This design enables doctors to retrieve a patient’s medical history in clinical practice and is also beneficial for us when obtaining patient-level data and then eye-level data. In our dataset, we treat left-eye and right-eye data from the same patient as two separate samples, as eye diseases may not occur in both eyes simultaneously. If I understand correctly, what you are concerned about is eye-level deduplication rather than patient-level deduplication.
>
> When processing the data, we assign a unique identifier to each eye. For example, the left eye of patient 1 is assigned eye1, the right eye of patient 1 is assigned eye2, the left eye of patient 2 is assigned eye3, and the right eye of patient 2 is assigned eye4. Then with our stratified sampling strategy, we can ensure that the training, validation, and test sets of $D_A$ are disjoint with respect to the eye index. This also holds for $D_B$.
>
> We denote the train/val/test sets of $D_A$ and $D_B$ as $Train_A$, $Val_A$ and $Test_A$, and $Train_B$, $Val_B$ and $Test_B$. We ensure
>
> $Train_A \cap (Val_A \cup Test_A \cup Val_B \cup Test_B) = null$
>
> $Train_B \cap (Val_A \cup Test_A \cup Val_B \cup Test_B) = null$
>
> $Val_A \cap (Train_A \cup Test_A \cup Train_B \cup Test_B) = null$
>
> $Val_B \cap (Train_A \cup Test_A \cup Train_B \cup Test_B) = null$
>
> $Test_A \cap (Train_A \cup Val_A \cup Train_B \cup Val_B) = null$
>
> $Test_B \cap (Train_A \cup Val_A \cup Train_B \cup Val_B) = null$
>
> This can avoid label information leakage.
>
> Tasks defined on $D_A$ and $D_B$ are different, thus data from a same eye can be included in both $Train_A$ and $Train_B$ or in both $Val_A$ and $Val_B$ or in both $Test_A$ and $Test_B$ so that we can fully make use of the data we collected for model training and evalution. That is to say, below claims hold:
>
> $Train_A \cap Train_B \neq null$
>
> $Val_A \cap Val_B \neq null$
>
> $Test_A \cap Test_B \neq null$
>
> We hope we have solved your concern and thank you very much again for the valuable discussion.

---

> > ### Comment · Reviewer_iVNK · 2025-11-28
> >
> > Thank you for your response. I currently have no further issues and will reconsider my rating.
> >
> > An additional suggestion would be to include more visualizations of different samples within the dataset in the appendix.

---

> > > ### Author Response · Authors · 2025-11-28
> > >
> > > Thank you very much for the discussion and we will include more visualizations of samples in the appendix.

---

### Author Response · Authors · 2025-11-12
**Could dot find reviewers' comments**

Dear PC, SAE and AE,

We could not find the reviewers' comments under the author' console page. Could you please help to set the reviewers' comments to be visible for us? Thank you very much!

---

> ### Author Response · Authors · 2025-11-12
>
> On the submission page, it shows “0 Official Reviews Submitted.” There may be an issue with the settings for this submission.

---

### Meta-Review · Area_Chair_Yrrk · 2026-01-08

**Summary:**

This paper introduces a mixed multimodal ophthalmic imaging dataset with diverse modality combinations (UWF-SLO and OCT images) for multi-retinal disease detection, along with PersonNet as a baseline model.

Key strengths identified across reviews:

- Well-organized presentation and clear motivation addressing a clinically relevant problem
- Seven real-world modality combinations reflecting personalized clinical workflows—a notable advancement over existing datasets with fixed modality pairs
- Substantial dataset size (8,558 eyes/5,235 patients) with disease prevalence aligned with epidemiological statistics
- Comprehensive benchmark evaluation of 13 existing methods alongside the proposed baseline
- Complete code and data release with proper ethical considerations

Primary concerns raised:

Label quality and reliability: Partial reliance on EMR records and treatment inferences, with some samples marked "unclear"; lack of inter-rater protocols (Reviewer iVNK)
Dataset scope limitations: Single hospital/vendor, limited to three diseases, lacking external validation (Reviewers nhTT, Zydr, bWtN)
Method novelty: PersonNet components (prototype-based completion, SKNet fusion) deemed straightforward; limited technical innovation (Reviewers nhTT, Zydr)
Evaluation gaps: Missing clinically critical metrics (ROC-AUC, calibration, subgroup analysis initially); ablation studies only on validation set (Reviewer iVNK)
Data split concerns: Insufficient clarity on patient-level deduplication and potential information leakage between benchmarks DA and DB (Reviewer iVNK)

**Reviewer Concerns:**

Adequately Addressed:

Data splits: Authors provided detailed mathematical notation demonstrating eye-level disjoint splits within DA/DB while allowing controlled overlap between benchmarks to maximize data utilization. The stratified sampling strategy ensures no label leakage.

Evaluation metrics: Authors added ROC-AUC results showing PersonNet's competitive performance. The justification for prioritizing F1-score and Kappa over ECE/Brier for imbalanced data is well-founded with concrete examples. Subgroup analyses were already in Appendix A.6.



Outstanding Concerns:

Label quality: While the authors justify marking samples as "unclear" when two ophthalmologists cannot confidently diagnose (especially for glaucoma), the lack of formal inter-rater agreement protocols and label noise auditing remains a methodological limitation. The exclusion of "unclear" labels from loss computation mitigates but does not eliminate this concern.

Method novelty: Authors correctly positioned this as a Dataset & Benchmark track submission where the primary contribution is the dataset itself, with PersonNet as a strong but simple baseline. However, the reviewer's concern about limited technical innovation in PersonNet remains valid from a methods perspective.

Multi-center diversity: Single-hospital limitation acknowledged but not addressed in current work. This significantly affects the dataset's clinical generalizability claims.

**Reviewer Scores:**

The authors provided comprehensive, professional responses that successfully addressed many technical concerns, particularly regarding data splits, evaluation metrics, and novelty claims relative to existing datasets. The repositioning as a Dataset & Benchmark track submission appropriately contextualizes the limited method novelty.

Remaining substantive concerns:
1) Method novelty: While acceptable for dataset track, PersonNet's contributions are incremental. ICRL is a technical conference, thus the technical contribution is important.
2) Label quality assurance: Absence of formal inter-rater protocols and noise auditing.
3) Single-center limitation: Significantly impacts generalizability claims despite being common in medical imaging datasets.
4) Incomplete evaluation: Missing generative baselines, limited sensitivity analysis.

---

### Decision · Program_Chairs · 2026-01-26

Reject